# Deep learning using electroencephalogram (EEG) data for diagnosing and predicting SSRI response in major depressive disorder

Sebastian Olbrich [1] ✉, Natalia Jaworska[2], Sara de la Salle[2], Verner Knott[2], Pierre Blier[2], Martin Brunovsky [3], Tobias Welt [1], Mateo de Bardeci[1] & Cheng Teng-Ip[4]

## Abstract

**Background** Major Depression (MDD) is a potentially life-threatening condition that ranks among the diseases with the highest global burden. Despite its prevalence, current diagnostic methods remain largely subjective, and first-line treatments exhibit high rates of non-responders.

**Methods** This study investigates the application of deep learning (DL) algorithms to electroencephalogram (EEG) data for the MDD-diagnosis and prediction of treatment outcomes following the administration of selective serotonin reuptake inhibitors (SSRIs), using six large, independent datasets with a total of n = 146 for healthy subjects and n = 203 for patients. DL models were trained on one portion of the datasets and tested on unseen data from different subjects. To interpret the classification features, Gradient-weighted Class Activation Mapping (Grad-CAM) was applied.

**Results** The models achieve an average accuracy of 67.5% (best fold 70%) in distinguishing MDD patients from healthy controls and mean 79% accuracy (best fold 85%) in predicting SSRI responders. Key EEG markers for both classification tasks revealed by Grad-CAM include alpha activity in the frontal and parietal regions. Simulation of a clinical decision scenario for SSRI treatment selection indicates a number needed to treat (NNT) of five when using a model with 80% predictive accuracy, corresponding to an increase in treatment response from a 50% baseline to 70% with model-guided selection. Conclusion: These findings underscore the clinical potential of EEG-based DL models for stratified treatment in MDD, facilitating accurate therapy choices and reducing ineffective treatments. The results of the integration of objective neurophysiological markers into clinical psychiatry are indicating the potential for more personalized treatment allocation.

## Plain Language Summary

This study looked for better ways to diagnose major depression and choose the right treatment. Today, depression is often diagnosed through interviews, and many people do not respond to the first medicine they receive. We tested whether computer models could learn patterns in brain activity (Electroencephalogram -EEG) to identify depression and predict who will benefit from a common antidepressant. We trained deep-learning models on large EEG datasets and tested them on new subjects. The models could detect depression well and even more accurately predict treatment success. These results show that EEG-based computer tools may help doctors choose treatments more accurately and reduce the time patients spend on ineffective therapies.

In the realm of physical medicine, encountering a somatic (i.e. non-psychiatric) symptom initiates a comprehensive diagnostic process to ascertain the optimal therapeutic intervention. This rigor contrasts starkly with the intervention approach in psychiatry, where individuals presenting with depressive symptoms rarely undergo an in-depth examination of the implicated organ, namely, the brain. Despite the significant personal toll and increasing prevalence of patients with major depression[1], our understanding of the brain's function in the context of a stratified medicine in psychiatry has remained stagnant for decades. As such, the central nervous system is still largely treated as a 'black box' when considering effective treatment options for depression[2].

Nonetheless, the landscape of psychiatric diagnostics and tailored treatments is not entirely devoid of innovation. Tools capable of monitoring cerebral activity with precise spatial and remarkable temporal resolution as

[1]University of Zurich, Psychiatric Hospital, Zurich, Switzerland. [2]University of Ottawa Institute of Mental Health Research at The Royal, Ottawa, ON, Canada. [3]National Institute of Mental Health, Klecany & Third Faculty of Medicine, Charles University Prague, Czech Republic. [4]Center for Cognitive and Brain Sciences, University of Macau, Taipa, Macau, SAR, China. ✉e-mail: sebastian.olbrich@pukzh.ch

well as widespread clinical availability already exist. The human electro-encephalogram (EEG), first recorded a century ago by Hans Berger[3], epitomizes a method that might be optimal for the assessment of brain function for diagnostic and predictive reasons. While its utility extends to diagnosing neurological conditions like epilepsy, assessing sleep disorders, and aiding anaesthesiologists in gauging sedation depth, its application within psychiatry has been curiously limited[4].

Recent shifts in clinical research foci have, however, heralded a renewed interest in the utility of EEG over other brain imaging modalities such as functional magnetic resonance imaging (fMRI) for its practicality in everyday psychiatric settings. Emerging studies highlight EEG's value, both in diagnosis and prognosis, particularly using conventional electrophysiological features captured during simple resting-state conditions in the field of psychiatry. Despite scepticism over the diagnostic relevance of some classical EEG markers, such as, theta/beta ratio for ADHD[5] diagnosis or frontal alpha asymmetry in depression[6], other EEG-derived markers have predicted treatment outcomes in patients with major depression[7] yet are not yet routinely used in practise. Good examples of this are EEG-vigilance markers for prediction of the response to selective serotonin reuptake inhibitors (SSRI) versus selective norepinephrine reuptake inhibitors (SNRI) in large cohorts[8]; this has been replicated in an independent sample[9]. The same is true for the utility of frontal alpha asymmetry[10], also independently replicated[11] for its prediction in drug response. However, the advent of machine learning algorithms in the field of prediction research has presented a paradox[12]: although it is possible to use such algorithms to surpass conventional markers in predictive accuracy, the interpretability of machine-learning and artificial intelligence (AI) findings remain elusive[13]. Specifically, crucial spatial, temporal or even causal insights from the EEG remain "trapped", that is, they are not fully exploited, within algorithmic confines.

The occurrence of deep learning and convolutional neural networks promised a breakthrough in addressing some of the above-mentioned limitation, yet their potential in EEG diagnostics and prognostics has been largely untapped. This might stem from a scarcity of sufficiently large and diverse datasets, hindering algorithmic refinement and validation across various contexts and recording configurations.

To address these limitations, the current study aims to evaluate the diagnostic and predictive capabilities of deep learning algorithms across six extensive and independent EEG datasets from different laboratories across the world in both healthy controls (HC) and patients with major depression (MDD) treated with the most frequently used first-line therapy in depression, namely the selective serotonin reuptake inhibitors[14,15] (SSRIs), such as citalopram and escitalopram. Our objective is not only to differentiate depressed and healthy (i.e., non-depressed subjects without a current psychiatric diagnosis) but also -and most critically- to predict treatment response. After all, response to SSRIs -which remain the first-line treatment of choice in patients with depression- hovers at around 50% and prolongs the "trial-and-error" approach in treating patients with MDD. Due to this prolonged suffering, it is critical to identify potential responders and non-responders to treatment classes based on baseline data of yet unmedicated or untreated patients. Achieving this could revolutionize patient care by enabling the selection of more effective initial treatments, thereby accelerating the journey to response and remission for many.

Additionally, this study seeks to reverse-engineer the deep learning models to decipher the EEG features underlying these classifications (i.e., HC vs. MDD patient; responder vs. non-responder). By translating these features into identifiable spatial and temporal patterns, we aim to illuminate further research avenues in this domain.

## Methods

### Response definition
Across all the sites, treatment response was defined as a > 50% decline in depressive symptoms as assessed with the rater-based questionnaire, either the Montgomery-Åsberg Depression Rating Scale (MADRS)[16], or the Hamilton Depression Rating Scale (HDRS)[17]. Duration until assessment of response was defined within the studies of the corresponding datasets and

varied between 4-8 weeks. Written informed consent to participate in the study was obtained from participants at all sites. Details are provided in Table 1. Ethical approval for this analysis was obtained from the faculty of medicine, University Zurich.

### CANBIND dataset
As part of the Canadian Biomarker Integration Network in Depression (CAN-BIND-1, details provided elsewhere[18]), eyes closed (EC) resting state EEG data (8 minutes) from $N = 309$ patients with MDD and $N = 146$ HC recorded between 2013 and 2017 were accessed through a controlled data release (https://doi.org/10.60955/mnwq-sq07). Participants were labelled with unique subject IDs, and all identifiable information was removed from the data files. All patients were treated with the SSRI escitalopram (10-20 mg/d) for 8-weeks; symptom severity was assessed at baseline and after 8 weeks using the MADRS. Data from $N = 175$ patients and 54 HC were used for EEG analysis. From initially 309 recorded datasets from patients at the CANBIND study, 212 sufficient EEG datasets from baseline were available for this analysis, with 37 datasets missing MADRS scores after 8 weeks, resulting in a total amount of 175 patient datasets to be included in the analysis for this study. This secondary data analysis study was approved by the Royal Ottawa Health Care Group Research Ethics Board.

### Ottawa dataset
As part of a larger clinical trial[19], 3-min of EC resting state EEG data from $N = 51$ unmedicated patients with MDD and $N = 43$ HC were recorded between 2008 and 2011. Patients were randomized (double-blind) to one of three antidepressant regimens: escitalopram + bupropion, escitalopram + placebo or bupropion + placebo. Only patients ($N = 19$) randomized to receive the SSRI escitalopram were included in the current analysis. Baseline symptom severity and outcome after 12 weeks were assessed using the MADRS. This study was approved by the Royal Ottawa Health Care Group Research Ethics Board.

### Leipzig dataset fMRI
As part of the LIFE-Adult project Leipzig[20], Germany EC resting state EEG data from $N = 31$ patients with MDD and $N = 32$ HC were recorded between 2012 and 2014. Patients were treated with different medications, in total, $N = 6$ received SSRIs were used for the predictive analysis. Baseline symptom severity and outcome after 4 weeks were assessed using the HRRS. This study was approved by the University Leipzig Ethics Board.

### Leipzig dataset prediction
As part of the Vigilance Algorithm Leipzig (VIGALL) project[21], Germany, EC resting state EEG data from $N = 22$ patients with MDD and 17 HC were recorded between 2013 and 2014. Patients were treated with different medications, in total, $N = 15$ received SSRIs and were used for the predictive analysis. Baseline symptom severity and outcome after 4-8 weeks were assessed using the Hamilton Depression Rating Scale (HDRS). This study was approved by the University Leipzig Ethics Board.

### Praha dataset 250 Hz
As part of the II-D-AD-QEEG project, Czech Republic, EC resting state EEG data from 67 patients with MDD were recorded between 2005 and 2011. Patients were treated with different medications, in total $N = 21$ received SSRIs and were used for the predictive analysis. Baseline symptom severity and outcome after 4-6 weeks were assessed using MADRS. This study was approved by the Ethics Board of Praha, Charles University.

### Praha dataset 1000 Hz
As part of ther research projects carried out at NIMH Klecany, Czech Republic, resting state EC EEG data from 93 patients with MDD were recorded between 2015 and 2019. Patients were treated with different medications, in total $N = 36$ received SSRIs and were used for the predictive analysis. Baseline symptom severity and outcome after 4-6 weeks were

**Table 1 | All included datasets with healthy controls (HC) and patients with major depression (MDD) with features of the clinical data and the EEG data**

| Dataset | HC (total/included) | MDD (total/included) | Sex | Responder (% of included) | Treatment | Assessment | Week of Assessment | Channel Number | Sampling Rate (Hz) | Duration EEG (HC/MDD) (sec) |
|---|---|---|---|---|---|---|---|---|---|---|
| Leipzig I | 32/32 | 31/6 | M 36% | 33% (n = 2) | Escitalopram | HDRS-17 | 4 | 27 | 1000 | 900/906 |
| Leipzig II | 17/17 | 22/15 | M 39% | 47% (n = 7) | Escitalopram (12), Citalopram (1) | HDRS-17 | 4 to 8 | 31 | 1000 | 127/126 |
| Prague I | 0 | 21/21 | M 24% | 62% (n = 13) | Sertralin (8), Fluvoxamin (2), Citalopram (1), Escitalopram (10) | MADRS | 4 to 6 | 19 | 250 | na/72 |
| Prague II | 0 | 36/36 | M 23% | 56% (n = 20) | Sertralin (14), Fluoxetin (4), Fluvoxamin (2), Citalopram (7), Escitalopram (5), Paroxetin (4) | MADRS | 4 to 6 | 19 | 1000 | na/75 |
| Ottawa | 43/43 | 19/14 | M 37% | 43% (n = 6) | Escitalopram | MADRS | 12 | 32 | 512 | 185/184 |
| CANBIND | 54/54 | 175/111 | M 37% | 50% (n = 55) | Escitalopram | MADRS | 8 | 58 | 512 | 555/562 |
| TOTAL | 146/146 | 309/203 | M 35% | 50.7% (n = 103) | SSRIs | HDRS or MADRS | 2 to 12 | 10 for all EEGs | 250 for all EEGs | 441/320 |

HC healthy controls, MDD major depressive disorder, Hz Hertz, sec seconds, M male, HDRS Hamilton Depression Rating Scale, MADRS Montgomery-Åsberg Depression Rating Scale.

assessed using the MADRS. This study was approved by the Ethics Board of Praha, Charles University.

### Preprocessing pipeline and standardization

Export to edf-format was done using Brain Vision Analyzer 2.2 (Gilching, Germany). All closed-eyes resting state EEG files were visually screened using DeepPSY software (version 1.02, Zollikerberg, Switzerland). Additional calculations of EEG-features, filtering and artefact removal were done using MNE version 1.2.0. The overlap of EEG channels between all studies was identified (10 channels: 'F7', 'F4', 'P3', 'O1', 'F3', 'C4', 'F8', 'O2', 'P4', 'C3') and only these ten channels were kept for all datasets and analyses. A detailed description of the electrode positions can be found in the supplement (Supplementary 1.1). Further, the lowest sampling rate was chosen (250 Hz) to standardize across all the EEG datasets and a low-pass filter (100 Hz) was applied before downsampling to avoid aliasing of frequency components. All EEG files then were filtered between 0.5 (high-pass cut-off for low frequency noise) and 45 Hz (low-pass cut-off to exclude muscle artefacts usually starting at 30 to 60Hz[22]) and segmented into 2 second segments after generating an average reference. We used a 0.5–45 Hz band-pass filter as a compromise to reduce low-frequency noise while retaining potential gamma-range EEG activity ( > 30 Hz), acknowledging in the limitations that residual muscle artifacts may persist in this range. (Additional analysis on a 0.5-30 Hz band pass filter can be found in the Supplementary 1.11). This resulted in a different number of segments for the patients and control subjects of the different datasets, ranging from 36 segments per subject to 450 segments. Additionally, we applied an eye-artefact correction method[23] based on electrooculogram EOG) data or reconstructed EOG channels (F7-F8) and a bad segment rejection method based on the min-max amplitude criterion ( ± 100 μV threshold) for deep learning and machine learning analysis. Further, we conducted a comparison of the amount of eye-related events/minute for all datasets (details in Supplementary 1.12). Details regarding the different datasets is outlined in Table 1. To achieve a higher generalization of the results for future research, a more detailed description of the recording settings from the different sites can be found in the supplement (Supplementary 1.7). Due to the strongly unbalanced number of EEGs across sites, enforcing uniform representation of all datasets in each batch would require extensive oversampling of smaller datasets, thereby skewing site weights and increasing the risk of overfitting rather than improving interpretability; thus, site-specific accuracies are provided in the supplement (Supplementary 1.8) instead.

### Deep learning models

Using a Keras Backend for TensorFlow (version 2.4.1) with Python 3.8 on a RTC 3090 GPU, the model used in previous work[24,25] was implemented for both differentiating (i.e., discrimination of HC and people with MDD) and predictive tasks (i.e., responder vs. non-responder). The Convolutional Neural Net (CNN) included six convolutional layers with 200 filters, employing kernel sizes from (2, 2) to (1, 2), followed by max pooling and dropout layers, with a dropout rate of 0.3 after the first five convolutional blocks and 0.5 after the sixth. Hyperparameters were adjusted using a random search approach. A hyperparameter grid, including the number of filters, dense layer size, dropout rate, and learning rate, was defined (see Supplementary 1.6). Using RandomizedSearchCV, we trained multiple configurations on the training data, evaluated them using cross-validation, and selected the best-performing model. The final model was trained on the optimized hyperparameters and evaluated on the test set.

The network was configured to receive inputs of shape (10, 500, 1) and outputs a binary classification. To minimize inter-dataset variability, all EEG segments were amplitude-normalized by subtracting the mean and dividing by the standard deviation of the training set prior to model training, with the same scaling applied to the test data. Training was conducted over 900 epochs with an Adamax optimizer set to a learning rate of $5 \times 10^{-5}$, and early stopping was implemented with a patience of 70 on validation loss to prevent overfitting with a total possible maximum duration of 700 epochs. Weight change was adjusted to fit the unbalanced datasets, resulting in a

**Fig. 1 | This diagram represents the structure of the best performing Deep Learning Network used for classification.** Each layer type is distinctly colored, transitioning from blue (early layers) to red (final layers). A bold vertical arrow on the left side emphasizes the top-to-bottom data flow, while the legend at the bottom-left corner provides clarity on each layer type (Conv2D Convolutional 2 Dimensional).

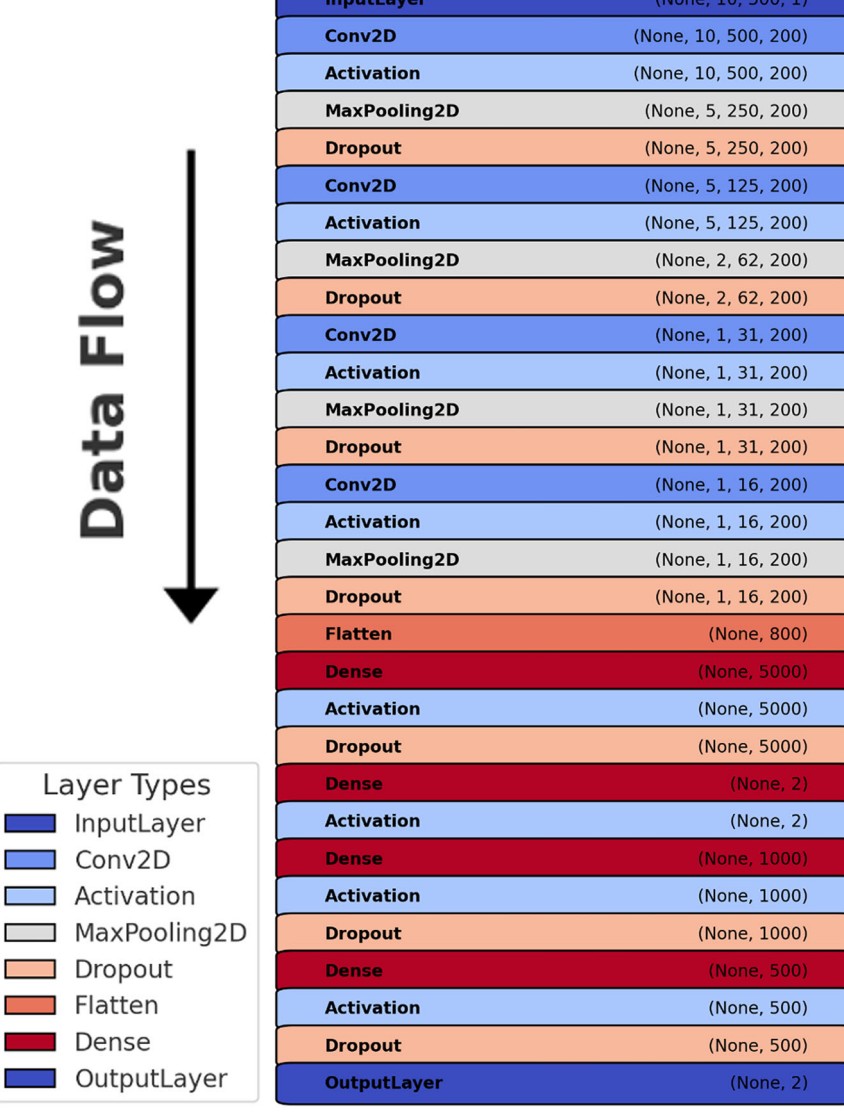

weight of 2:3 for HC versus MDD and 17:20 for Responders versus Non-Responders. A listing of all convolutional network layers can be found in Fig. 1.

**Statistics and reproducibility: training, validating and testing**

The EEG files suitable for further analysis from all datasets (CANBIND; Prague I/II, Leipzig I/II and Canada) were separated into a training set, a validation set and a test set. "Files" here refers to the complete recording of one subject while "segments" refer to a 2-second segment of on subject. The test set consisted only of EEG files from patients or controls that had not been used for training or validation to make sure the network did not learn individual features. This is a very important step of the testing procedure since it guarantees the generalizability of the results toward unseen patients and controls. The test set was randomly chosen with 40 EEG files set aside for classification test set MDD versus HC (balanced 20 files each group) and 20 files set aside for classification test set Responder versus Non-Responder (balanced 10 files each group). The files were randomly distributed across the recording sites.

Within the remaining corpus of EEG files, 5% of the segmented files (all files were split into 2 s segments) were used for validation using a random shuffle and splitting routine. To prevent shuffling segments from one subject into the training and testing sets, all segments received identifiers according to their subjects of origin. An early stopping of leaning epochs was implemented following a failed decrease of the validation loss of the optimizer function for consecutive 70 epochs. The maximum of epochs for training was set to 900, but never was used since validation loss criteria for early stopping were fulfilled earlier in all trials. It is important to note, due to an overlap of terminology in EEG research and Deep Learning research, that, in this paper, an epoch is not an EEG segment, but a run of the convolutional model through all training data. A short period of timeseries data from the whole EEG is called a segment instead (i.e., 2 s segment).

It is also important to note that the main outcome was set to the correct labelling of subjects, not single segments. That means, a subject in the test set was counted as correctly classified if 50% or more of single segments were classified according to the appropriate label of the subject (i.e., patient or control; responder or non-responder). Thus, there were different accuracies for single segment classification and for subject-wise classification. Although this approach could over- or underestimate the accuracy of the model at the segment-level, it seems to be the more meaningful approach since it is clinically relevant to identify subjects (e.g. responders), and not an EEG-segment.

**Grad cam and important EEG topographies and frequencies**

The method employs Gradient-weighted Class Activation Mapping (Grad-CAM) to visualize areas of interest in the 2D EEG arrays that influence the predictions of the CNN. By averaging the activation maps from

convolutional layers, it generates heatmaps that highlight the significant regions for the model's decisions. The code snippet iterates over a set of images, computing Grad-CAM heatmaps for each, then overlays these averaged heatmaps onto the original images for visualization. Finally, it computes and displays an overall averaged heatmap across a subset of images to identify common patterns of activation. This approach is instrumental in interpreting the model's behavior by revealing which features contribute most to its predictions. We binarized normalized Grad-CAM maps at $\tau = 0.3$ to generate seeds, following common practice in WSSS (Weakly Supervised Semantic Segmentation)/CAM pipelines, and prior imaging work that reports/visualizes Grad-CAM at 0.3[26]. We then calculate a Fourier spectrum for every marked sequence from every two-second segment. Thus, a power spectrum is obtained for every channel that highlights the EEG frequencies with the largest discriminative power. Further, the visualisation of the weighted Grad-CAM outputs allows for the comparison of the diagnostic and predictive tasks by means of topography.

### Conventional machine learning algorithms

For comparison of the differentiation between MDD and HC as well as for the differentiation between Responders and Non-Responders to SSRI treatment, several conventional EEG-parameters that have been found to be important for diagnostic or predictive purposes in MDD[11,27,28] were computed for all subjects after artefact—and eye movement removal. This included EEG-power of the Delta (0.5-4 Hz), Theta (4-8 Hz), Alpha (8-12 Hz), Beta (12-20 Hz) and Gamma (20-45 Hz) frequency bins at all 10 electrodes ('F7', 'F4', 'P3', 'O1', 'F3', 'C4', 'F8', 'O2', 'P4', 'C3' used for the combined analysis, Alpha peak frequency and averaged lagged linear connectivity measures between frontal, parietal, central and occipital lobes (total 6 variables), resulting in a total of 47 variables. The EEG-variable-matrices were then used for a machine learning (ML) approach using ten different established ML algorithms: In the past, EEG-based classification utilized various machine learning algorithms, each with distinct strengths. Linear models like Logistic Regression and Naïve Bayes offer interpretable decision-making but may struggle with complex patterns. Tree-based methods such as Decision Trees, Random Forest, Gradient Boosting, LightGBM, and XGBoost enhance accuracy through ensembling, with Random Forest and XGBoost being particularly robust for EEG feature differentiation. Distance and margin-based classifiers, including K-Nearest Neighbors (KNN) and Support Vector Machines (SVM), are commonly used for tasks like motor imagery classification and seizure detection. Boosting techniques like AdaBoost and Gradient Boosting iteratively improve weak classifiers but may be sensitive to noise. Thus, we implemented these ten algorithms with a "leave-one-out" cross-validation approach to compare the results to the deep-learning approach.

### Usage of other deep learning architectures (EEGNet)

To compare the presented results with the performance with other state of the art deep learning networks, we used the EEGnet design to compute a model based on the same input data for the comparison between Responders and Non-Responders. The structure of the model was derived from the original work[29]. EEGNet is a lightweight deep convolutional neural network (CNN) designed specifically for EEG signal classification tasks such as brain-computer interfaces (BCIs), motor imagery, and seizure detection. It employs depthwise and separable convolutions to efficiently learn both spatial and temporal EEG features, reducing computational complexity while maintaining high performance. The network typically consists of three main layers: (1) a temporal convolutional layer (2) depthwise convolution to capture spatial patterns across EEG channels, and (3) a separable convolution. Details can be found in the supplement (Supplemenary 1.4).

## Results

### MDD versus HC

Classification of MDD subjects versus HC subjects suitable for further analysis from all datasets (CANBIND; Prague I/II, Leipzig I/II and Canada) showed a train accuracy of 73.16% and a validation accuracy of 71.13% (see

Fig. 2, top right). Test accuracy for all pooled segments of unseen subjects from the test set was 64.84% (SD 2.0%) in total (not subject-wise) and the percentage of correctly classified unseen subjects based on the number of correct classified segments of a subject was 67.5% (average for 5-fold validation with best fold 70%, SD 1.8%) with a standard deviation of 30.4% across subjects. The confusion matrix yielded a sensitivity of 68.4% and a specificity of 66.7% (see Fig. 3, left).

When using an eye-artefact reduction method[23] and exclusion of artefact segment using a min-max approach, results were worse than without extensive preprocessing (correctly classified unseen subjects 65%, SD 28.4%). No differences in eye-related movements were found between groups (Supplementary 1.12). A slightly decreased accuracy was also found when applying a 30 Hz low-pass filter instead of 45 Hz low pass filter (see Supplementary 1.11) with 65%.

### MDD SSRI prediction

Classification of responders versus non-responders of SSRI treatment showed an average of 71.8% accuracy for single segments for the training set after 732 training epochs. Validation set accuracy was 66.4%. The test set in these trials reached an average of 64% (SD 4.2%) for pooled segments and a mean value of 79% (5-fold average, best fold 85%, SD 2.5%) for correct subject-wise classification. The confusion matrix yields a sensitivity of 75% and a specificity of 87.5% with a standard deviation of 24.7% across subjects (see Fig. 3, right).

Again, when using extensive preprocessing procedures, the results were found inferior with correctly classified unseen subjects 70%, SD 21.0%). Regarding the fact that different times until response were used in the different datasets, we decided to do a sensitivity analysis. Thus, the accuracies for correct labelling were compared for responders and non-responders in dependence of the timepoint of the MADRS or HDRS assessment. There was no significant difference between sites with assessments from weeks eight to twelve (Canada and CANBIND) and other sites with assessments between weeks four to eight (t-statistic: −0.84, $p > .41$). A decreased accuracy of 55% was also found when applying a 30 Hz low-pass filter instead of 45 Hz low pass filter (see Supplementary 1.11). No differences in eye-related movements were found between responders and non-responders (Supplementary 1.12).

### Machine learning classification and EEGNet results

The results of the ten different machine learning results for the differentiation of patients with MDD and HC varied between 52.1% (Naïve Bayes) to 60.6% (XGBoost) and 66.1% (Random Forest). Thus, the results of the tree-based algorithms came close to the results of the Deep Learning approach but did not reach the same accuracy. Regarding the differentiation between Responders and Non-Responders, the results of the ML approaches stayed way behind the accuracy of the Deep Learning approach with some algorithms staying below random-guess accuracy (e.g. Gradient Boosting with 47.1%) to a maximum of 60.3% with Ada Boost. More details can be found in the supplement (Supplementary 1.3).

Although the computing time was much lower in comparison to the other DL approach, the results of the EEGNet (more details on the model can be found in the Supplementary 1.4 and 1.5) for differentiation between Responders and Non-Responders stayed well below the already reported results with training and validation accuracies reaching 74% each, but achieving finally only 55% of correctly classified subjects (SD 31.6%) in the test set. Among the tested architectures, the final CNN achieved the best performance due to optimized hyperparameters, appropriate regularization, and its capacity to capture the complexity of EEG patterns, rather than model depth alone.

### Grad-CAM, EEG topographies and frequencies

Using the Grad-CAM technique, the layer activations for the subjects in the unseen test set were identified at the segment-wise subject level as well as the averaged global level over all subjects. This allowed us to map the topography of EEG-channel importance for the label classification (Figs. 2 and 4,

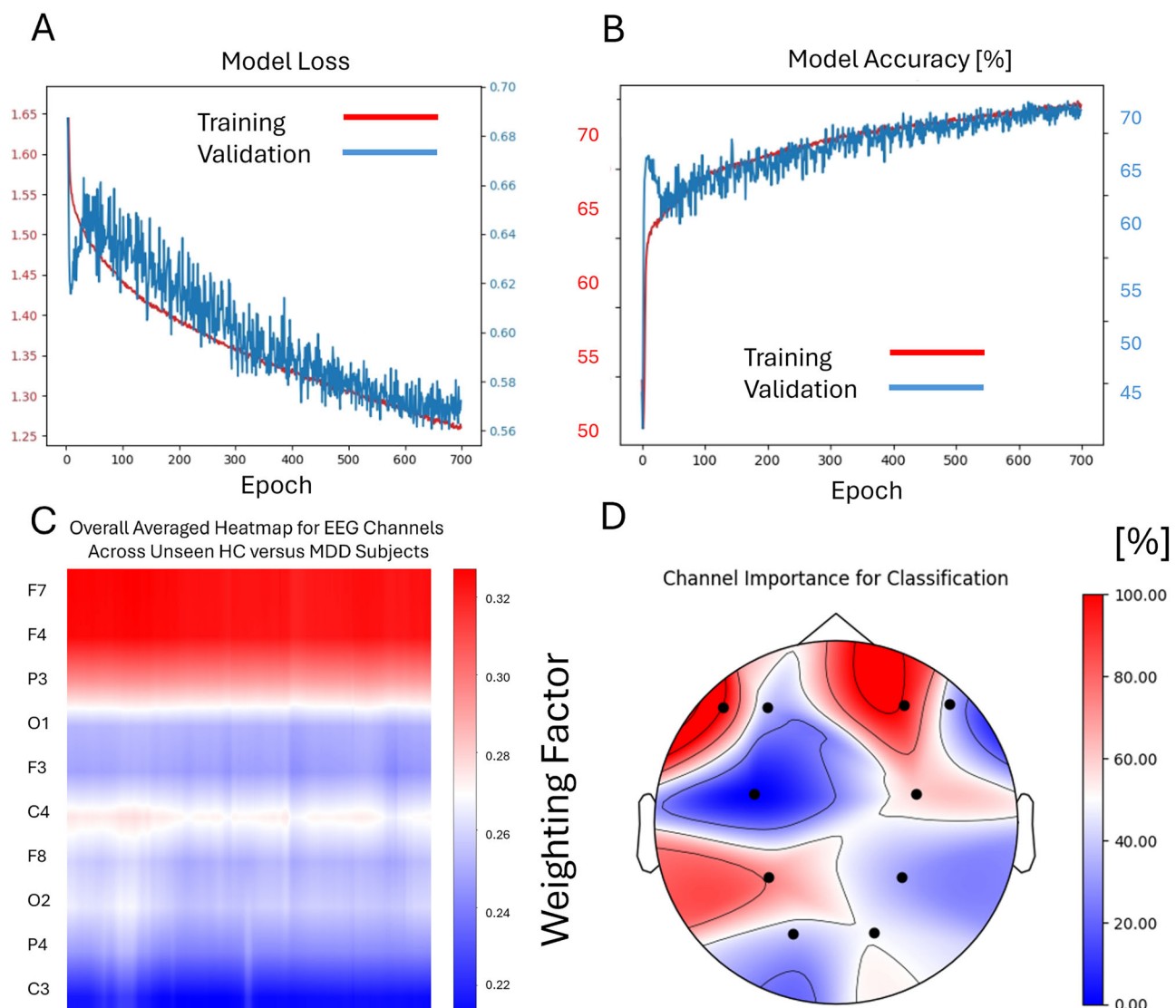

**Fig. 2 | Illustration of the training and validation process of the Deep Learning model for the discrimination of patients with Major Depression (MDD, *n* = 203) and Healthy Controls (HC, *n* = 146).** While the loss of the model declined throughout the epochs (top left, **A**), the accuracy for discrimination increased (top right, **B**). Using the Grad-CAM method, the averaged importance of each channel for the test dataset was illustrated (bottom left, **C**), showing higher values for frontal F7, F4 and the parietal P3 electrodes (bottom right, **D**).

bottom). The resulting maps for the HC versus MDD classification showed (Fig. 2) a clear accentuation of the electrodes at F7, F3, P3 and C4 while electrodes O1, O2, F3, F8, P4 and C3 showed less involvement in the classification process. Results of mapping these results onto the scalp-electrode positions, the relevant topography for EEG activity that divides HC subjects from patients with MDD, can be seen in Fig. 2, bottom right.

Using the same technique for the differentiation between Responders and Non-Responders, it became apparent that only electrodes at F3 and F7 showed a high weight in correct classification of patients. The mapping results can be seen in Fig. 4, bottom right.

Following this, we tried to determine the frequencies of EEG activity that contributed most to the correct classification. We therefore computed heatmap overlays for all classified EEG segments (example Fig. 5). Then, we extracted the periods with a weight >0.3 for all channels and calculated the EEG power spectra (see Fig. 6). The most prominent peak for all relevant channels can be found between 8-12 Hz, resembling the EEG alpha rhythm.

**Number-needed-to-treat**

Using the confusion matrix for treatment response with an accuracy of 80%, we identified that 9 patients are true positives and 7 are true negatives. When the marker is applied, responders (9 patients) receive SSRI treatment, achieving a 100% response rate. Non-responders (7 true negatives and 3 false negatives) receive SNRI treatment, which has a 50% response rate, resulting in 5 responders. In total, 14 out of 20 patients respond to treatment, yielding an overall response rate of 70%. This represents a 20% improvement compared to the baseline response rate of 50% for SSRI treatment without the marker. Table 2 depicts the increase of responders and decrease of numbers needed to treat for a rising accuracy of correct classification. Under the assumption that SSRIs and SNRIs as alternative show a 50% response rate in an unselected patient cohort[30], an accuracy of 80% in the prediction of SSRI response using the EEG has a number-needed-to-treat (NNT) of five. We added the receiver-operator-characteristics and the area-under-the-curve values in the supplement (Supplementary 1.10). To make the results comparable to other studies, we computed also the normalized positive predictive value (nPPV) as:

$$nPPV = \frac{PPV}{Baseline\ Response\ Rate}$$

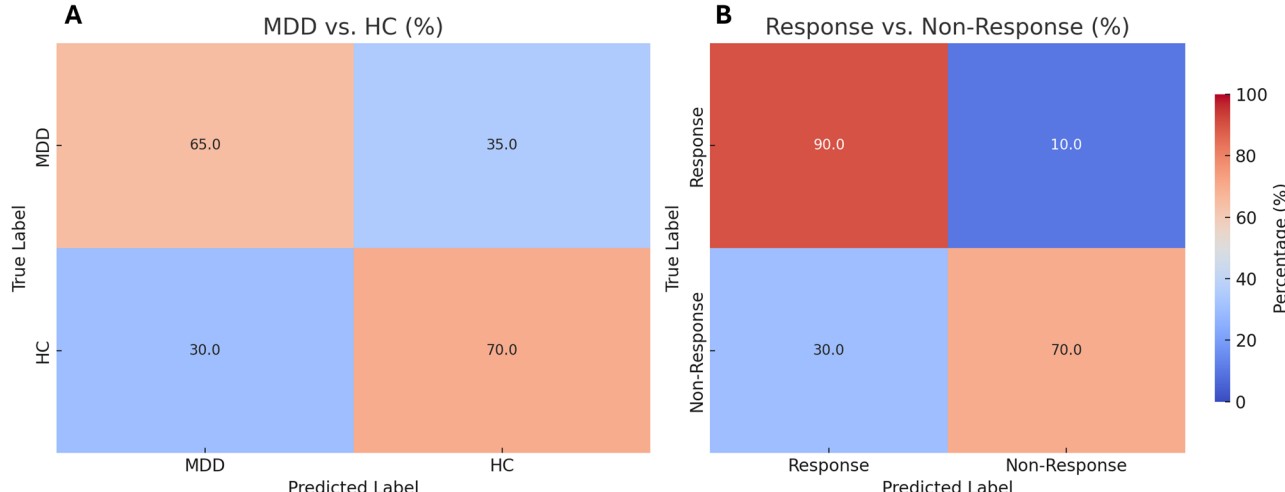

**Fig. 3 | Confusion matrices for classification of patients with Major Depression (MDD) versus Healthy Controls (HC) and for classification of Responders versus Non-Responders.** Panel **A** shows differentiation accuracies between patients and healthy controls. Although the model at panel **B** shows slightly higher true positive values for responders (90%) than for non-responders (70%), this difference is moderate and clinically acceptable; false negative predictions do not harm patients, as they still receive alternative treatments, while the improved detection of responders substantially enhances treatment allocation.

## Discussion

In brief, in this study, a robust deep learning model was trained to correctly identify two thirds of MDD subjects and HC in an unseen (i.e., completely new) part of the dataset from different laboratories across the world using a relatively low sampling rate (250 Hz) and only 10 electrodes under eyes-closed conditions of various length. Most notably, with an even higher mean accuracy of 79%, another trained model correctly predicted the treatment outcome following SSRI medication (i.e., responder versus non-responder), again using separate subject datasets from different laboratories. These results outperformed traditional machine learning algorithms, especially in the prediction task. Looking closer into the discriminative features, the Grad Cam retrospective activation of the deep learning network revealed a specific pattern of EEG activity to be discriminative for diagnostic and predictive purposes, namely, frontal alpha activity was a marker of SSRI treatment response.

While many psychiatrists may rate the fairly good discriminative power for diagnostic purposes rather unimportant, attributing to the specialized physician's ability to diagnose depression, this finding might be notable from the viewpoint of general practitioners. While depression is among the leading causes of disability worldwide and most patients with depression are seen by a general practitioner as the first contact with the medical system, around 50% of cases remain undiscovered or are diagnosed incorrectly[31]. Thus, enhancing diagnostic accuracy in the general practise might have utility. However, since the presented cohort might be more severely depressed than patients general practitioners typically see (i.e., the patients were all tested at psychiatric hospitals or research institutes), the usability of an EEG assisted diagnosis in a broader/community-based population has to be further validated. Although many EEG deep-learning studies report very high diagnostic accuracies, these often result from methodological biases such as data leakage or small sample sizes, leading to overestimation[32]. Given that inter-rater reliability for MDD diagnosis among clinicians is only poor to moderate ($\kappa = 0.28$–$0.60$)[33], our more modest accuracy likely reflects a realistic estimate under clinically valid conditions.

Even more important and relevant for a psychiatrist or general practitioner is the fact that the DL model was able to predict response and non-response for SSRI use with a very high average accuracy of 79%. While other EEG based AI models have been shown to predict similar aspects of treatment decisions[12], this is the first to use EEGs from different laboratories with only a small subset of electrodes and relatively low sampling rate, thus making it attractive and feasible for a large array of settings in out- and inpatient populations. In other words, this approach could lead to an expedited diagnosis and guidance regarding further specialized follow-up. However, using only a set of two frontal electrodes, namely the F3 and F4, yielded only minimally better accuracy than a random choice model (not shown). Thus, the optimal number of channels, also for future home and smartphone-based EEG recording devices might be somewhere between three and ten channels. As a limitation of the current results, the usage of an average reference in a ten-channel montage might have influenced the findings; thus, this issue must be addressed in future work. Furthermore, including both eye-artifact reduction and min-max artifact exclusion notably reduced sample size, and classification accuracy declined in both HC vs. MDD and responder vs. non-responder comparisons—suggesting reduced statistical power due to data loss rather than artifact-driven classification. Indeed, recent literature shows that traditional artifact rejection does not reliably improve deep-learning performance—in some cases it offers no benefit or even worsens classification outcomes[34]. To rule out potential confounding effects of group differences in eye-movement–related events, we quantified their occurrence across all settings and found no significant differences between patients and healthy controls, nor between responders and non-responders (see Supplementary 1.12). Moreover, applying a more stringent low-pass filter at 30 Hz instead of 45 Hz resulted in reduced classification accuracies across settings. It remains unclear whether this decrease reflects the removal of informative but potentially muscle-related electrophysiological activity, or whether genuine EEG gamma-band activity in the 30–45 Hz range contributes to the improved performance observed when retaining these higher frequencies. The results further show that standard deviation between different runs of a model were small (1.8%-4.2%) while inter-subject standard deviation for classification results was much larger (21.0%-28.4%). This implies that although the models seem to be robust, there still might be some space to more accurately assess individual features with improved deep-learning models.

Limitations noted, the findings of this study are valuable clinically, since the deep learning approach with a training set, a validation set and a held-out unseen testing set resamples in parts an external replication needed for the validation of a biomarker. Thus, the high accuracy for discriminating responders from non-responders that survived the training, validation and -finally- the testing phase, coupled with the fact that the recordings in all three sets came from different laboratories from around the world (i.e., different amplifier systems and recording conditions; although all shared 10 electrodes and the EC resting state), indicates that these findings are robust and might be generalizable. The EEG seems to measure a robust biological

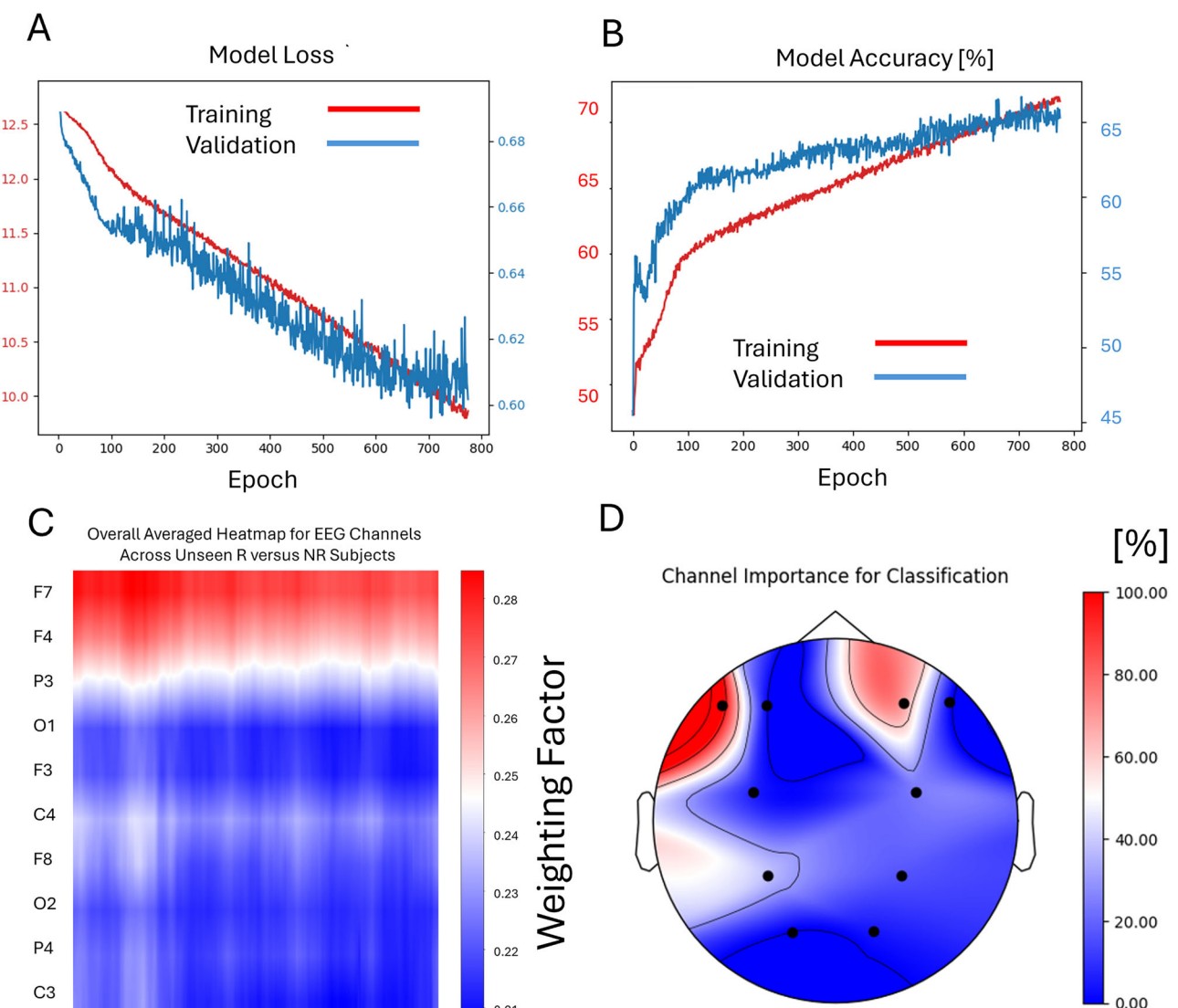

**Fig. 4 | Illustration of the training and validation process of the Deep Learning model for the discrimination of Responders and Non-Responders to SSRI treatment (total *n* = 203).** While the loss of the model declined throughout the epochs (top left, **A**), the accuracy for discrimination increased (top right, **B**). Using the Grad-CAM method, the averaged importance of each channel for the test dataset was illustrated (bottom left, **C**), showing higher values for frontal F7, F4 electrodes (bottom right, **D**).

signal at a relatively low-technology level, rendering the resulting time-series quite comparable across different setting. This makes EEG a potentially ideal tool for clinical use, and contrasts with recently failed replications of fMRI results for biotyping depression[35].

The graphical visualization of the importance weighting of the different channels reveals clear patterns, with the F4 electrode (frontal right cortex) being the most significant for classifying responders and non-responders, as well as for diagnostic labelling. At both the individual and segment levels of classification importance (Fig. 5), and when examining the power spectra of the decisive EEG frequencies, the model weights are focused on EEG alpha activity. Previous research has transitioned from viewing alpha asymmetry diagnostically to considering it predictively[6,10]. However, prior studies and replications found that frontal alpha asymmetry was only relevant for predicting SSRI response in female subjects[10,11]. Due to the limited amount of data, a separate analysis on sex-specific predictivity could not be performed in this work and should be focus of future approaches. Still, the model's effectiveness for a mixed sex cohort may be explained due to the revealed Grad-CAM pattern, which also assigns high importance to additional channels, specifically the F7 electrode at the lateral frontal left cortical area and the P7 electrode at the left parietal cortex. Bruder et al.[36]

demonstrated that alpha asymmetry at less frontal sites could help identify male responders to SSRIs. It is possible that the deep learning model identified and integrated these markers to optimize predictive power.

The importance of alpha activity in the frontal and parietal regions also suggests the involvement of the default mode network. However, a more detailed understanding of the sources responsible for correct classification would require source localization techniques. According to the principles of these localization algorithms, using only 10 channels of EEG is insufficient to identify plausible intracortical sources. To achieve this, a large dataset with approximately 200 responders and non-responders to a single treatment, such as SSRIs, and an EEG electrode recording set of more than 19 channels would be necessary.

The accuracy of predicting SSRI responders and non-responders holds significant clinical, social, and economic importance. Depression is one of the most prevalent and potentially deadly disorders worldwide, and SSRIs are the most commonly prescribed medication for depressive symptoms[37,38]. However, treatment guidelines indicate that each treatment trial without an objective marker takes two to six weeks before a change in strategy is considered[39–41]. This delay until response is necessary in almost half of the cases, as the response rate to first-line SSRI treatment is only 50%[30].

Following this, many patients switch to another medication, often an SNRI, or augment their treatment with additional drugs such as aripiprazole or lithium, continuing the cycle of 50% non-response.

Our results demonstrate that an EEG biomarker capable of differentiating between responders and non-responders with an accuracy of almost 80% could increase the number of first-line responders to 70%, resulting in a number needed to treat (NNT) of five. In comparison, the NNT for SSRIs themselves, calculated from a large meta-analysis, is also five[42]. This underscores that the clinical use of EEG markers would represent a substantial improvement in the management of depression. When using the normalized positive predictive value for e.g. the accuracy of 70% used in this work, the NNT even drops to 2.5, further highlighting the potential of this method.

From a technical standpoint, it remains unclear if the arrangement of the channels in the input matrix or the duration of the EEG recording influenced the accuracy of the model. Due to computing power limitations, we did not test whether changing the order of electrodes in the 10×500

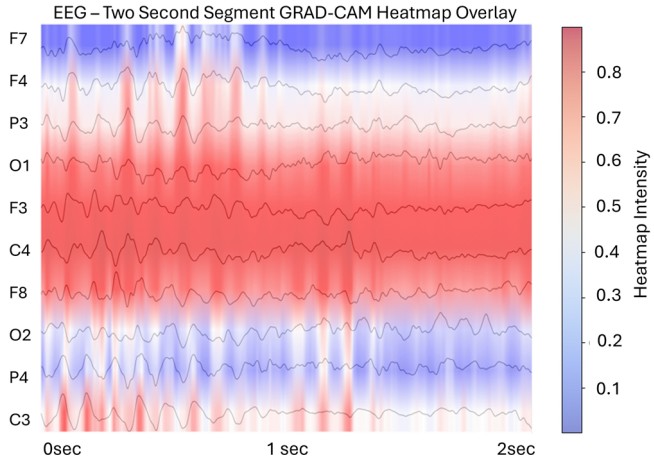

**Fig. 5 | Illustration of the overlay of an averaged heatmap derived from all six Deep Learning layers over a single two-second EEG segment.** Important features are marked red, less important features are blue. EEG alpha activity seems to be in the focus of highly activated weights, especially at channels F3, P3 and C3 at the given example.

(electrodes x timesteps) input matrices would have affected accuracy. It is possible that reordering the channels to improve spatial information could enhance accuracy by allowing the model to combine adjacent features more effectively[32]. Additionally, using different reference montages, instead of calculating an average, could impact the results; however, an average montage affords most flexibility. These aspects should be investigated further in comparable future work.

Regarding the duration of the EEG recording, this is important from two perspectives. First, brain states change during the resting state, leading to gradual shifts in brain activity from wakefulness to sleep onset[43,44]. It is unclear if the features distinguishing responders from non-responders depend on EEG vigilance[8,9], and if maintaining a specific wakefulness level could enhance predictive power. Second, more available segments can improve subject-wise prediction accuracy[25]. Therefore, a balance must be struck between obtaining enough segments for each individual and avoiding overly long recordings that may cause the brain to transition into less distinctive functional stages.

A limitation for this study arises from the fact of the assessment of response at different points in time for the different datasets. Although we tried to standardize the data as much as possible and a sensitivity analysis showed no differences in the results for different timepoints of assessment, these differences might still have affected the results and lowered the accuracy of the models. Thus, for future studies, a more standardized approach to assessment of response is needed, potentially following a more unified approach for clinical evaluation. Another limitation can be found in the fact, that comorbidities were not included into the computational model. Although all studies reported depression as main diagnosis, larger datasets are needed to shed light on the influence of psychiatric comorbidities on the results.

## Conclusion

This study demonstrates that a biomarker-based approach for treatment allocation, specifically using deep learning networks with low-cost EEG data, could significantly enhance psychiatric care. If implemented clinically, this approach could streamline treatment allocation, reducing the need for long and costly development pipelines and trials. In conclusion, this study proves the utility of deep learning networks for utilizing EEG data to accurately allocate SSRI treatment. Large-scale clinical implementation of these models could proceed without extensive trials, as the risk is minimal

**Fig. 6 | Display of the logarithmic EEG power spectra for all channels, averaged for all test patients, with a high importance for the correct classification for patients with Major Depression (MDD) versus Healthy Controls (HC), left panel, and for Responders (R) versus Non-Responders (NR), right panel.** The largest peak for almost all channels can be found in the EEG alpha frequency range from 8 to 12 Hz, indicating that this frequency is most pertinent for diagnostic and predictive classification. (Note: black horizontal lines give the y-axis height of $10^{-3}$ V²/Hz, red vertical lines denote the EEG-alpha range in the channels with the largest weights for classification).

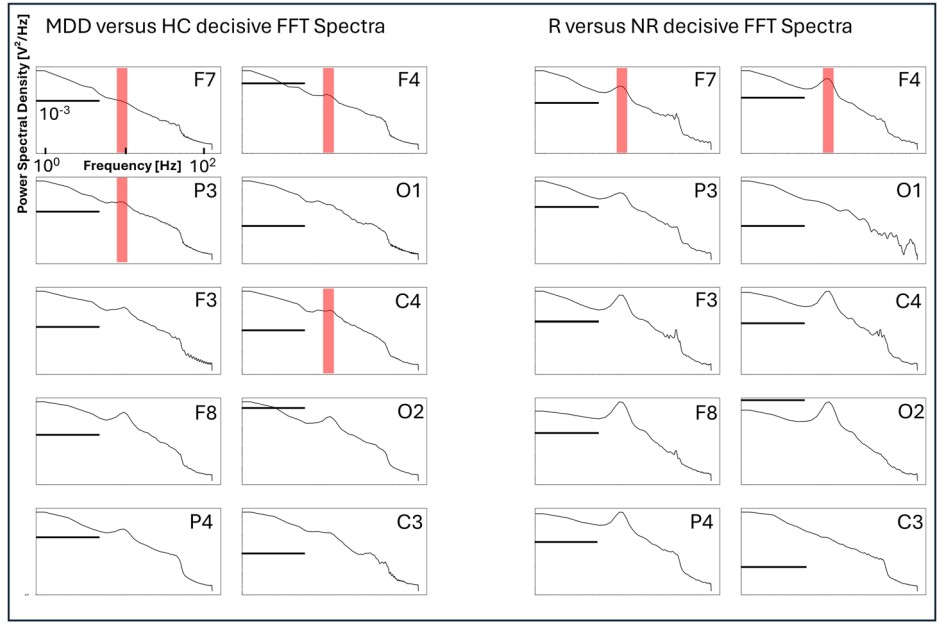

**Table 2 | Number-needed-to-treat (NNT) for scenarios with different treatment-response accuracies**

| Metric | Accuracy 70% | Accuracy 75% | Accuracy 80% |
|---|---|---|---|
| SSRI Responders Identified | 35% | 37.5% | 40% |
| SSRI Non-Responders Identified | 65% | 62.5% | 60% |
| SNRI Responders (50% of non-responders) | 32.5% | 31.25% | 30% |
| Total Response Rate | 67.5% | 68.75% | 70% |
| Improvement Over Baseline Response (50%) | 17.5% | 18.75% | 20% |
| Number Needed to Treat (NNT) | 5.71 | 5.33 | 5 |
| Sensitivity | 75% | 76.92% | 78.95% |
| Specificity | 87.5% | 88.24% | 88.89% |
| Positive Predictive Value (PPV) | 90% | 90.48% | 90.91% |
| Negative Predictive Value (NPV) | 70% | 71.43% | 72.73% |
| Normalized Positive Predictive Value (nPPV) | 116.67% | 125% | 133.33% |

An accuracy of 80% like in the presented cohort would lead to a 20% increase of the response rate with a NNT of 5 (SSRI selective serotonin reuptake inhibitor; SNRI serotonin norepinephrine reuptake inhibitor).

for already approved drugs, offering a promising shortcut toward more effective and efficient psychiatric treatment.

## Data availability

The data that support the findings of this study are available on request from the corresponding author S.O. The data are not publicly available due to them containing information that could compromise research participant privacy/consent. The source data for Fig. 2 is in Supplementary Data 1 and in Supplementary Data 2, the source data for Fig. 4 is in Supplementary Data 2 and in Supplementary Data 3, the source data for Fig. 5 is in Supplementary Data 4.

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

## Acknowledgements

We would like to acknowledge the individuals and organizations that have made Data used for this research available including the Canadian Biomarker Integration Network for Depression (CAN-BIND), the Ontario Brain Institute, the Brain-CODE platform, the Government of Ontario, as well as all collaborators listed in this link [www.canbind.ca/DataReleaseCBN01]. CTI was supported by the University of Macau (File no: SRG2023-00040-ICI & MYRG-GRG2024-00022-ICI) and the Science and Technology Development Fund (FDCT) of Macau (0092/2025/ITP2).

## Author contributions

S.O. designed the study, performed the analyses, and wrote the manuscript. N.J., S.d.l.S., V.K., P.B., M.B., T.W., and M.d.B. provided data, contributed to analyses, and reviewed the manuscript. C.T.-I. conducted additional analyses and reviewed the manuscript.

## Competing interests

S.O. is Co-Founder of DeepPSY company and served as consultant for several pharmaco-companies. All other authors declare no competing interests.
