## [Transparent Peer Review file · Communications Medicine]

Deep Learning Using Electroencephalographic (EEG) Data for Diagnosing and Predicting SSRI Response in Major Depressive Disorder

Corresponding Author: Professor Sebastian Olbrich

Version 0:

Reviewer comments:

Reviewer #1

(Remarks to the Author)

This study explores the application of a deep learning (DL) algorithm for analyzing EEG data to both diagnose Major Depressive Disorder (MDD) and predict treatment responses to SSRIs. Using data from six international datasets and 10 EEG channels, the authors developed a model that achieved a 67.5% accuracy in detecting MDD patients from healthy controls and an 80% accuracy in identifying responders to SSRI treatment from non-responders. While the study demonstrates promising results in utilizing deep learning for predicting SSRI treatment responses in MDD patients, several critical limitations should be considered to enhance the reliability and applicability of the analysis:

Major Concerns:

1. Lack of Comparison with Conventional Machine Learning Methods:

The study does not evaluate traditional machine learning (ML) algorithms alongside applied DL model. Given the extensive body of previous research using conventional ML methods with often higher accuracy for detecting MDD, the added value of employing deep learning in this context is unclear. It would be more insightful to compare the performance of the applied DL model against traditional ML techniques using the same dataset, as DL models are typically more computationally intensive. Moreover, only a single DL model architecture was tested without comparisons to alternative architectures. Exploring a range of DL models could provide insights into robustness and potential improvements in performance.

2. Limited Hyperparameter Optimization:

There is minimal discussion on the exploration and optimization of hyperparameters. This could limit the model's performance and its potential to achieve higher accuracy.

3. Inconsistent Treatment Durations for Response Assessment:

The treatment durations for assessing response vary significantly (ranging from 4 to 12 weeks). This inconsistency could introduce variability in response assessment, potentially affecting the model's accuracy and generalizability.

4. Inconsistent Recording Protocols Across Sites:

The variation in EEG recording protocols and devices across different sites is not adequately addressed. Standardizing recording protocols is essential for reducing variability and improving the reliability of model predictions.

5. Overlooked Comorbid Conditions:

The study does not take into account potential comorbid conditions that may affect EEG signals, which could lead to confounding results and reduce the model's diagnostic accuracy.

6. Threshold for Correct Classification in Segments:

The study states, "a subject in the test set was counted as correctly classified if 50% or more of single segments were classified according to the appropriate label of the subject (i.e., patient or control; responder or non-responder)." This 50% threshold may overestimate the model's performance. Higher thresholds can be checked, or it would be informative to report additional metrics such as the percentage of segments agree with classification result or the consistency of predictions

across segments or other metrics.

7. Arbitrary Threshold in Grad-CAM Analysis:

The study employs an arbitrary threshold of 0.3 for Grad-CAM analysis without providing justification. Clarifying the rationale behind this choice would help in understanding its impact on interpretability.

8. Insufficient Preprocessing and Artifact Removal:

The study lacks a systematic approach to artifact removal prior to analysis.

Other Concerns:

9. Limited demographic representation

10. Limited Information on EEG Recording Setup: There is insufficient detail about the EEG recording setups used across different sites, such as the devices and specific configurations, which could influence data quality.

11. Figure 1 lacks clarity. Improving both the visual presentation and the accompanying caption would enhance the readability.

Despite my major concerns about the paper, there are several positive aspects worth highlighting:

- The study's focus on reducing the burden of ineffective treatment trials is valuable, given the significant challenges in managing Major Depressive Disorder. Developing a predictive tool for SSRI response may help in personalizing treatment and improving patient outcomes.
 - The use of Gradient-weighted Class Activation Mapping (Grad-CAM) is a strength, as it enhances the interpretability of the deep learning model's results. This approach helps in identifying the specific EEG features contributing to the predictions, which is often a limitation of applying neural networks, especially DL models.
 - Incorporation of Multiple Datasets with Fewer Channels: The use of multiple datasets with a reduced number of EEG channels is a practical and noteworthy approach. However, while this strategy is promising, there are several issues in the current implementation that need to be addressed to fully realize its potential.
- In summary, while the paper demonstrates an innovative use of deep learning for EEG data analysis with some promising aspects, the lack of comparisons with conventional ML methods, unstandardized data collection protocols across centers (varying recording durations, different assessment timeframes, and inconsistent treatment periods), and limited exploration of model architectures significantly undermine its contributions.

Reviewer #2

(Remarks to the Author)

This is an important and timely manuscript that investigated use of AI/DL to diagnose MDD and especially to predict treatment response, and also use Grad-CAM techniques to further elucidate what the AI picked up. The study used strong methodology, including out-of-sample validation, and results are realistic with relatively low accuracy for diagnostic purposes and an 80% accuracy for treatment response, which is on line with the existing literature.

I have no major comments, just some minor comments:

- What explains the dataloss in CANBIND from N=308 to N=175 included?
- I was a bit surprised that only 10 EEG channels were shared between all datasets. The lowest channel dataset had 19 channels, and I would have expected more of these 19 channels, especially midline channels, would have been shared by datasets. Any comments on this?
- Could you also report the normalised PPV (nPPV) value, since that statistic becomes more often used, also in the light of future stratification options? I think this is what you tried to do under number needed to treat, but didn't name it as such, maybe rely there on terms such as PPV and NPV and calculate nPPV as the improvement of response as a result of utilizing the biomarker.
- Relatedly, I think the authors don't interpret the prediction data well. In the discussion they state: '...could increase the number of first-line responders to 70%, resulting in a number needed to treat (NNT) of five...' Yet the reported PPV is >90%. The Positive Predictive Value (PPV) indicates the number of responders when the biomarker would have been applied prospectively, so that would have been say 90%. The nPPV would thus be $90\%/70\%=129\%$. Maybe the NNT will thus also be higher?
- The authors argue that alpha seems to be the most promising feature, however, from figure 5 I can't really see that it is alpha, since some clear alpha bursts are marked blue. Maybe reconsider the example, or orient people to what they need to look at (could it be the throughs of alpha only?).
- The prominence of alpha and F4 for treatment prediction begs the question, what if all analyses would be repeated in females only, excluding males, would the results have been stronger, given prior reports have found alpha asymmetry to be a sex-specific predictor.

Some typo's:

Page 12: resampling should be resembling

Page 11: figure x needs updating.

Page 15: consider changing: ...keep-away unseen testing set.. Held-out? or Out-of-sample validation?

Reviewer #3

(Remarks to the Author)

In this study, Olbrich et al. used convolutional deep learning networks and EEG from multiple sites to classify patients with major depression vs. healthy controls and responders vs. non-responders to SSRI treatment. Results ranged from ~67-80%, promising further exploration. Although the proposed methodology and results are of interest to the field of computational psychiatry, the paper needs more clarity regarding some choices that reduce my excitement.

> Overall language:

There is an increasing effort in the field of psychiatry to use language that reduces stigma (please see <https://pmc.ncbi.nlm.nih.gov/articles/PMC8580983/>). Therefore, instead of using “depressed patients” (a disease-centered classification), I suggest using “patients with depression” as it supports a person-centered language.

> Response Definition:

I am concerned with the definition of response. The study mentions that “Duration until assessment of response was defined within the studies of the corresponding datasets and varied between 4-8 weeks.” This definition means that non-responders from one study (e.g., following 4 weeks of treatment) could be responders in another trial (e.g., following treatment for 8 weeks). These differences could interfere with the algorithm learning and inference. Why not standardize the response data across studies (e.g., response at 4 weeks) as done with the EEG data (e.g., selecting the minimum frequency and overlapping channels)?

> Datasets:

- Please clarify the rationale for including only part of the CANBIND dataset in the analysis.
- The text gives the impression that only patients from the Ottawa and Leipzig I/II datasets were included in the analysis, which differs from the information listed in Table 1. Please review the text to clarify this.

> EEG signal processing:

- Please list the software or toolboxes used to preprocess the EEG data.
- Please provide more details regarding the filter used for the EEG data and the rationale for the 0.5-45Hz interval.
- Are there any considerations regarding the potential effects of EOG/EMG artifacts? Even with eyes closed, eye movements or eyelid contractions could explain the frontal patterns shown in Figures 2 and 4.

> Training, Validating, and Testing:

- I assume “files” refer to all epochs from one single patient. If so, please clarify in the text.
- When it says that “5% of the segmented files (all files were split into 2s 184 segments) were used for validation using a random shuffle and splitting routine”, how did you ensure you were not including segments from the same participant in the training and validation sets?

> Results:

- If results show data from multiple epochs and the percentage of segments accurately classified across subjects (as described in the Methods section), please provide the standard deviations of these results.
- In the discussion, the text mentions, “using only a set of two 310 frontal electrodes, namely the F3 and F4, yielded only minimally better accuracy than a random choice model (not shown). Thus, the optimal number of channels, also for future home and smartphone-based EEG recording devices might be somewhere between three and ten channels.” If these findings support your discussion, they should be included, even if only as a supplement.

> Figures:

- Please include the resolution and dimensions of Figure 1, as it's barely readable in the current configuration.
- Please check Figure 2C, as the “C” textbox covers part of the graph title.
- What are the measurement units in the charts in Figures 2B and 4B? I assume Figure 2B shows percentages. If so, shouldn't both y-axis have the same scale (e.g., 50-80% or .5-.8)? The same applies to Figure 4.
- For Figure 3, both matrices could use percentage values and the same color bars to facilitate interpretation.
- In Figure 5, the EEG time segments are not correctly aligned with the labels on the left. Also, the signal from C3 seems predominantly out of the image.
- Figure 6: please specify in the caption what the red-shaded areas mean and what the black horizontal bars mean. Also, the black bar for P4 seems misplaced outside of the chart. Additionally, the acronym (Non-R) in the figure doesn't match the acronym NR in the caption.

> Please double-check if the dimensions listed in this sentence are correct: “The network is configured to receive inputs of shape (10, 500, 1)”.

> Check for typos. Examples:

- Line 63: “(...) EEG-vigilance markers for prediction of the prediction of selective serotonin (...)”
- Line 127: “projec21t”
- Line 314: “(...) thus, this issue must be addressed in future work..”

Version 1:

Reviewer comments:

Reviewer #2

(Remarks to the Author)

The authors implemented the feedback well except one thing, the calculation of the normalised PPV was incorrect, this should be calculated by: $PPV / (\text{Total sample Response Rate})$. So if a PPV of 80% is found and overall response rate is 60% then the nPPV is $80/60=133,33\%$, which can be interpreted that the response rate improved with 33,33% after applying the biomarker. A value below 100% (that the authors currently included) thus implicates the biomarker has NO added value.

Reviewer #3

(Remarks to the Author)

The authors have addressed most of my previous concerns; however, I still have reservations regarding the EEG preprocessing pipeline, which may significantly impact the validity of the reported classification performance.

1. Band-pass filter setting:

The manuscript states:

“All EEG files then were filtered between 0.5 (high pass cut off for low frequency noise) and 45 Hz (to exclude muscle artefacts usually starting at 30 to 60 Hz²²).”

However, filtering the data with a 45 Hz low-pass cutoff still allows a considerable portion of EMG-related activity (typically present from ~30 Hz up, as stated by the authors) to remain in the signal. Therefore, the filtering step does not fully mitigate muscle artifact contamination, which could inflate classification performance.

2. Eye movement artifacts and preprocessing trade-off:

The paper writes:

“When using an eye-artefact reduction method²⁵ and exclusion of artefact segment using a min-max approach, results were worse than without extensive preprocessing (correctly classified unseen subjects 65%, SD 28.4%).”

This raises a critical concern. It is well-documented that individuals with depression exhibit different eye movement patterns compared to healthy controls (e.g., Li et al. 2016, Shanghai Arch Psychiatry; Carvalho et al. 2015, Front Psychiatry; Takahashi et al. 2021, Front Psychiatry; Zhang et al. 2022, J Affect Disord). By not properly removing EOG-related activity, the model may be capitalizing on eye movement differences rather than neural EEG features. This undermines the interpretation that the classifier is operating solely on brain-based EEG data.

While including EOG-related features in a multimodal classification framework is valid, if the intention is to demonstrate EEG-based classification, then non-neural sources such as eye movements should be carefully identified and removed using established preprocessing pipelines. Otherwise, the results may reflect artifact-driven rather than brain-driven group differences.

Reviewer #4

(Remarks to the Author)

This is a review of the research manuscript titled 'Deep Learning Using Electroencephalographic (EEG) Data for Diagnosing and Predicting SSRI Response in Major Depressive Disorder' by Sebastian Olbrich et al. Since this is a revised submission, I appreciate that the authors have addressed most of the concerns raised during the previous review round. However, a few key issues still remain that need attention before the manuscript can be considered for publication.

1. Low Diagnostic Accuracy for MDD:

The reported diagnostic accuracy for Major Depressive Disorder (MDD) using the proposed deep learning (DL) model is relatively low. Existing literature demonstrates significantly higher accuracy using both traditional machine learning and deep learning approaches. This gap raises questions regarding model effectiveness and justifies further investigation.

2. Justification for Model Architecture and Performance:

The authors claim that the chosen architecture is the “deepest” among the ones tested, and hence performs better. I respectfully disagree with this rationale. Model depth alone does not guarantee better performance—factors like overfitting, data preprocessing, and regularization play critical roles. A deeper model may underperform if not properly tuned or if the dataset is insufficient or imbalanced.

3. Unclear Early Stopping Criteria:

The manuscript lacks a clear explanation of the early stopping criteria used during training. From the training curves provided, it appears that the model was still learning and had not yet overfitted, yet training was prematurely halted. Authors should clearly define the early stopping mechanism (e.g., validation loss plateau, patience parameter) and justify why training was stopped at that point.

4. Lack of Cross-Validation for Final Model Evaluation:

Although 3-fold cross-validation was mentioned for hyperparameter tuning in the supplementary material, no cross-validation was applied during the final model evaluation. Given the relatively low accuracy, I strongly recommend performing a more robust evaluation using 5-fold or 10-fold cross-validation and report individual and mean \pm SD accuracy of the model. This will help assess the generalizability and stability of the model across different data partitions.

5. Potential Dataset Bias and Amplitude Normalization:

The use of multiple datasets with potentially varying recording conditions might be a key factor contributing to low model accuracy. While frequency normalization and channel unification were mentioned, it is unclear whether amplitude normalization was also performed across datasets. I suggest the authors clarify this point and, if not already done, apply amplitude normalization to reduce inter-dataset variability.

6. Need for Dataset-Wise Performance Evaluation:

To further investigate the source of low accuracy, I recommend reorganizing the training data such that each batch includes samples from all datasets. Then, evaluate and report performance separately on test sets from each individual dataset. This would help identify whether performance degradation is uniform across datasets or driven by specific ones, aiding in result interpretation and potential improvement.

7. Bias in the Model:

Figure 3B's confusion matrix suggests that the model is biased toward the 'Response' class. This issue needs to be addressed. For example, if the 'Response' class achieves 100% detection while the 'Non-response' class achieves only 50%, the overall accuracy would still be 75%. However, such a model would not be practical or clinically useful. The authors should adjust their model to mitigate this bias.

8. Acronyms:

The authors have redefined several acronyms multiple times throughout the text and figure captions. It is recommended that each acronym be defined only once, upon first use, and then used consistently thereafter. Additionally, "MDD" is defined inconsistently—both as Major Depressive Disorder and Major Depression. The authors should choose one definition and apply it consistently throughout the manuscript.

Version 2:

Reviewer comments:

Reviewer #1

(Remarks to the Author)

Comments have been implemented well.

Reviewer #2

(Remarks to the Author)

I appreciate the authors' efforts to address my concerns regarding preprocessing choices and artifact handling. However, I remain unconvinced that the reported classification performance can be attributed primarily to neural EEG features.

Specifically, the decision to retain EMG-contaminated frequencies and to forgo proper rejection/correction of EOG artifacts willingly leaves the results vulnerable to contamination. The argument that reduced performance after artifact removal is due to data loss, rather than muscular/ocular confounds, remains speculative. Adding a citation that artifact rejection can sometimes harm deep-learning classification is interesting, but it does not specifically address disorders known to have systematic EMG/EOG confounds, such as depression. Given the black-box nature of deep learning models, it is not possible to determine empirically from the current work whether the classifier is relying on neural EEG activity or non-neural signals (EOG/EMG). While the authors have added statements to the limitations, these do not resolve the central concern that the findings may be artifact-driven.

For these reasons, I do not feel comfortable endorsing publication of the manuscript in its current form. I defer the final decision to the editor.

Reviewer #3

(Remarks to the Author)

Thank you for addressing my concerns thoroughly. While I partially disagree with the authors' justification that all studies reporting high accuracy are prone to data leakage and subject bias, I do agree that small datasets are a concern. Therefore, the justification provided is valid. I have no further issues with this manuscript.

Reviewer #4

(Remarks to the Author)

This manuscript presents a compelling study that applies deep learning techniques to EEG data for both the diagnosis of major depressive disorder and the prediction of SSRI treatment response. The topic is clinically significant, and the inclusion

of multiple independent datasets enhances the potential generalizability of the findings. The approach of combining deep learning with Grad-CAM interpretability adds methodological novelty. Nevertheless, several methodological clarifications, consistency issues, and minor stylistic adjustments are needed before the work can be considered for publication :

Title

The title uses Electroencephalographic Data, whereas the abstract and main text alternate between electroencephalography data and electroencephalogram. Although these terms emphasize slightly different aspects, consistency in terminology throughout the manuscript is strongly recommended.

Abstract

- 1.It is suggested to briefly indicate the total sample size to enhance the credibility of the study.
- 2.The statement regarding a “number-needed-to-treat (NNT) of five” is a notable point for clinical translation. However, please clarify the basis of this estimate—specifically, which model accuracy (e.g., 79%) and what assumptions (e.g., a 50% response rate for SNRI treatment) were used in the calculation.
- 3.The expression “paving the way for optimized treatment strategies” is somewhat promotional. A more academic phrasing such as “indicating the potential for more personalized treatment allocation” would be preferable.

Introduction

The introductory analogy comparing psychiatry to a “broken arm” in physical medicine appears somewhat unnecessary and distracts from the central topic. It is recommended to remove this passage and focus more directly on the context of psychiatric disorders.

Methods

- 1.The study integrates data from six public datasets, which could introduce systematic bias due to heterogeneity in factors such as sex, age, ethnicity, assessment time points, and medication types. Please clarify in the Methods or Results how such confounding effects were controlled (e.g., through sensitivity analyses or weighting strategies).
- 2.The analysis was restricted to ten overlapping EEG channels across the six datasets. The authors should specify whether these ten electrodes represent exactly the same scalp locations in all datasets, as different studies follow varying EEG montages (e.g., 10–10 vs. 10–20 systems).
- 3.The EEG preprocessing pipeline appears non-standard. The manuscript mentions downsampling to 250 Hz prior to band-pass filtering (0.5–45 Hz), which may cause frequency aliasing and potential signal distortion.
- 4.If the gamma band is retained, the authors should justify its physiological relevance in this study or discuss possible contamination from muscle artifacts.
- 5.Data imbalance was addressed only by weighting adjustments. Please indicate whether other methods such as SMOTE or class weighting were tested as alternative approaches.
- 6.The choice of the Grad-CAM threshold (0.3) lacks justification. Providing a sensitivity analysis or referencing relevant literature would strengthen the methodological rigor.

Results

- 1.It is recommended to include ROC curves or report AUC values as complementary performance metrics, providing a more robust assessment of model accuracy.
- 2.Several results show large standard deviations (around 30%), suggesting substantial interindividual variability or data instability. Please elaborate on these observations in the Results or Discussion sections.
- 3.The reporting of SD should be consistent throughout the manuscript. In several revised sections, the percentage symbol (%) was omitted after SD values.

Conclusion

The conclusion section is overly lengthy and could be condensed to emphasize the key findings and implications of the study more clearly.

Reviewer #5

(Remarks to the Author)

Version 3:

Reviewer comments:

Reviewer #4

(Remarks to the Author)

The author has made a lot of improvements to address all the reviewers' issues and has basically adopted my suggestions. Therefore, I recommend the acceptance and publication of this manuscript.

Reviewer #5

(Remarks to the Author)

I believe the authors have sufficiently addressed the remaining concern.

Dear reviewers,

the authors are grateful to the reviewer's critical comments and thoughtful suggestions. Based on the comments we received; we revised the whole manuscript, added information and new calculations were necessary.. All changes were marked in red text in the manuscript. The reviewers concerns are marked **bold**, the answers in normal and the added text for the manuscripts are *cursive*. Please find our point-to-point replies below.

Reviewer #1

This study explores the application of a deep learning (DL) algorithm for analyzing EEG data to both diagnose Major Depressive Disorder (MDD) and predict treatment responses to SSRIs. Using data from six international datasets and 10 EEG channels, the authors developed a model that achieved a 67.5% accuracy in detecting MDD patients from healthy controls and an 80% accuracy in identifying responders to SSRI treatment from non-responders. While the study demonstrates promising results in utilizing deep learning for predicting SSRI treatment responses in MDD patients, several critical limitations should be considered to enhance the reliability and applicability of the analysis:

Major Concerns:

R1.1

Lack of Comparison with Conventional Machine Learning Methods:

The study does not evaluate traditional machine learning (ML) algorithms alongside applied DL model. Given the extensive body of previous research using conventional ML methods with often higher accuracy for detecting MDD, the added value of employing deep learning in this context is unclear. It would be more insightful to compare the performance of the applied DL model against traditional ML techniques using the same dataset, as DL models are typically more computationally intensive.

Moreover, only a single DL model architecture was tested without comparisons to alternative architectures. Exploring a range of DL models could provide insights into robustness and potential improvements in performance.

We thank reviewer 1 for this important concern. We decided to follow the advice of reviewer #1 and add important additional analysis to the already reported analysis of our work. Thus, we computed machine learning algorithm-based results for the differentiation of patients with depression and healthy controls as well as for the differentiation of responders and non-responders. Firstly, we applied automatic artefact detection methods to all EEG datasets, including bad segment detections and EOG correction. Secondly, we extracted several EEG features (based on literature findings) that are known to be useful in the identification of patients with depression and in the differentiation between SSRI responders and non-responders, including EEG alpha power asymmetry, alpha-peak frequency, EEG power spectra of the main frequency bands (theta, alpha, beta and gamma band) and also connectivity measures between main brain regions for every subject. Thirdly, we applied ten different machine learning algorithms to the feature matrix (Logistic Regression, K-Nearest Neighbours (KNN), XGBoost, LightGBM, Naive Bayes, SVM, Decision Tree, Random Forest, Gradient Boosting, AdaBoost). The description of these methods and the results for both scenarios (HC versus MDD and R versus NR) are given in the passages below. We included new passages in the main text and also gave more extensive descriptions in the supplement (supplement 1.1 and 1.3).

“Conventional Machine Learning Algorithms

For comparison of the differentiation between patients with depression and healthy controls as well as for the differentiation between Responders and Non-Responders to SSRI treatment, several conventional EEG-parameters that have been found to be important for diagnostic or predictive purposes in MDD^{11,26,27} were computed for all subjects after artefact - and eye movement removal. This included EEG-power of the Delta (0.5-4Hz), Theta (4-8Hz), Alpha (8-12Hz), Beta (12-20Hz) and Gamma (20-45Hz) frequency bins at all 10 electrodes ('F7', 'F4', 'P3', 'O1', 'F3', 'C4', 'F8', 'O2', 'P4', 'C3' used for the combined analysis, Alpha peak frequency and averaged lagged linear connectivity measures between frontal, parietal, central and occipital lobes (total 6 variables), resulting in a total of 47 variables. The EEG-variable- matrices were then used for a machine learning (ML) approach using ten different established ML algorithms: In the past, EEG-based classification utilized various machine learning algorithms, each with distinct strengths. Linear models like Logistic Regression and Naïve Bayes offer interpretable decision-making but may struggle with complex patterns. Tree-based methods such as Decision Trees, Random Forest, Gradient Boosting, LightGBM, and XGBoost enhance accuracy through ensembling, with Random Forest and XGBoost being particularly robust for EEG feature differentiation. Distance and margin-based classifiers, including K-Nearest Neighbors (KNN) and Support Vector Machines (SVM), are commonly used for tasks like motor imagery classification and seizure detection. Boosting techniques like AdaBoost and Gradient Boosting iteratively improve weak classifiers but may be sensitive to noise. Thus, we implemented these ten algorithms with a “leave-one-out” cross-validation approach to compare the results to the deep-learning approach.”

For the differentiation of healthy subject and patients with depression, some ML algorithms came close to the performance of the DL network, but eventually not reaching the same accuracy. For the prediction of treatment response, the ML algorithms did by far not reach the accuracy of the DL network.

“Machine Learning Classification and EEGnet Results

The results of the ten different machine learning results for the differentiation of patients with depression and healthy controls varied between 52.1% (Naïve Bayes) to 60.6% (XGBoost) and 66.1% (Random Forest). Thus, the results of the tree-based algorithms came close to the results of the Deep Learning approach but did not reach the same accuracy. Regarding the differentiation between Responders and Non-Responders, the results of the ML approaches stayed way behind the accuracy of the Deep Learning approach with some algorithms staying below random-guess accuracy (e.g. Gradient Boosting with 47.1%) to a maximum of 60.3% with Ada Boost. More details can be found in the supplement.”

And added a line to the end of the following paragraph in the discussion section:

“[...] These results outperformed traditional machine learning algorithms, especially in the prediction task. “

Additionally, we added this to the supplement 1.3:

Machine Learning Algorithms and results of ML classification

1. AdaBoost (Adaptive Boosting)

- *Type: Ensemble learning (Boosting)*
- *Features: Iteratively improves weak classifiers; sensitive to noise; works well with structured data but prone to overfitting on noisy datasets.*

2. Decision Tree

- *Type: Tree-based model*

- *Features: Simple, interpretable; prone to overfitting; splits data hierarchically for classification.*
3. Gradient Boosting
- *Type: Ensemble learning (Boosting)*
 - *Features: Strong learner built from weak classifiers; powerful but computationally intensive; good for capturing complex patterns.*
4. K-Nearest Neighbors (KNN)
- *Type: Distance-based (Instance-based learning)*
 - *Features: Non-parametric; classifies based on proximity to neighboring data points; sensitive to feature scaling and high-dimensional data.*
5. LightGBM
- *Type: Gradient boosting framework*
 - *Features: Optimized for large datasets; computationally efficient; fast training while maintaining high accuracy.*
6. Logistic Regression
- *Type: Statistical model*
 - *Features: Linear classifier; probabilistic output; interpretable; works well for simple decision boundaries.*
7. Naïve Bayes
- *Type: Probabilistic classifier*
 - *Features: Assumes feature independence; computationally efficient; effective in text and medical classification tasks.*
8. Random Forest
- *Type: Ensemble learning (Bagging)*
 - *Features: Multiple decision trees reduce overfitting; robust and generalizable; handles non-linearity well.*
9. Support Vector Machine (SVM)
- *Type: Margin-based classifier*
 - *Features: Finds the optimal hyperplane for classification; effective in high-dimensional spaces; supports non-linear classification via kernels.*
10. XGBoost (Extreme Gradient Boosting)
- *Type: Optimized gradient boosting*
 - *Features: Fast training; built-in regularization to prevent overfitting; widely used in competition-grade ML applications.*

Results of classification

Algorithm	MDD vs HC [%]	Responders vs Non-Responders [%]
AdaBoost	0.5685	0.6029
Decision Tree	0.5308	0.4803
Gradient Boosting	0.5753	0.4705
KNN	0.6164	0.5
LightGBM	0.6404	0.4509
Logistic Regression	0.5685	0.5245
Naive Bayes	0.5205	0.4852
Random Forest	0.6609	0.446
SVM	0.589	0.5245
XGBoost	0.6061	0.4313

Concerning the critic of only a single net being used, we want to apologize missing to report more extensively how we decided to use this network. The network itself is deeper than other DL networks

used commonly and the main difference between this network and e.g. the EEGnet network is that it is convoluted within time and channels at the same time, uses much higher number of kernels and adds a much larger dens layer. The network we present has been used in other EEG labelling tasks by our group with good results. However, to be transparent, we now add the comparison to an EEGnet approach and report the findings in the main text as well as in the supplement.

Supplement:

“1. Temporal Convolution Block (Learning Frequency Filters)

- *Conv2D (1,64, 16 filters, no bias, same padding): Applies temporal convolutions across the EEG time dimension to learn frequency-specific features.*
- *Batch Normalization: Normalizes the activations, improving stability and convergence.*

2. Depthwise Convolution Block (Learning Spatial Features)

- *DepthwiseConv2D (10,1, depth multiplier=2): Applies spatial convolutions, learning EEG channel-wise dependencies while keeping the number of parameters low.*
- *Batch Normalization: Normalizes feature maps.*
- *Activation (ELU): Adds non-linearity to improve feature extraction.*
- *Average Pooling (1,4): Reduces spatial dimensions, retaining key information.*
- *Dropout (0.5): Prevents overfitting by randomly deactivating neurons.*

3. Separable Convolution Block (Time-Frequency Refinement)

- *SeparableConv2D (1,16, 32 filters, same padding): Further refines extracted features by applying pointwise and depthwise convolutions.*
- *Batch Normalization: Maintains stable activations.*
- *Activation (ELU): Adds non-linearity.*
- *Average Pooling (1,8): Further reduces dimensions for efficiency.*
- *Dropout (0.5): Additional regularization.*

4. Fully Connected Classification Layers

- *Flatten: Converts feature maps into a vector for classification.*
- *Dense (64 Neurons, ELU): A fully connected layer for feature representation.*
- *Dropout (0.5): Further reduces overfitting.*
- *Dense (2 Neurons, Softmax Output): The final layer classifying EEG data into two categories.*

Model Summary

- *Total Parameters: 34,114*
- *Trainable Parameters: 33,954*
- *Non-trainable Parameters: 160*

Layer (type)	Output Shape	Param #
Conv2D	(None, 10, 500, 16)	1024
BatchNormalization	(None, 10, 500, 16)	64
DepthwiseConv2D	(None, 1, 500, 32)	320
BatchNormalization	(None, 1, 500, 32)	128
Activation	(None, 1, 500, 32)	0

AveragePooling2D	(None, 1, 125, 32)	0
Dropout	(None, 1, 125, 32)	0
SeparableConv2D	(None, 1, 125, 32)	1536
BatchNormalization	(None, 1, 125, 32)	128
Activation	(None, 1, 125, 32)	0
AveragePooling2D	(None, 1, 15, 32)	0
Dropout	(None, 1, 15, 32)	0
Flatten	(None, 480)	0
Dense	(None, 64)	30784
Dropout	(None, 64)	0
Dense	(None, 2)	130
Total Parameters		34114
Trainable Parameters		33954
Non-trainable Parameters		160

And more in the supplement about the Results of the EEGNet classification (supplement 1.5)

Results of the EEGNet with training and validation accuracies and losses (note that the results have been truncated at epoch 50)”

Main Text:

“Although the computing time was much lower in comparison to the other DL approach, the results of the EEGNet (more details on the model can be found in the supplement) for differentiation between Responders and Non-Responders stayed well below the already reported results with 55% of correctly classified subjects (SD 31.6%).”

By means of the tuning of the hyperparameters, please refer to section R1.2 below.

R1.2

Limited Hyperparameter Optimization:

There is minimal discussion on the exploration and optimization of hyperparameters. This could limit the model's performance and its potential to achieve higher accuracy.

Again, this is a very important remark and we are glad of having the chance to describe our procedure to more detail. The network we use is a deep network with a very high number of kernels and a very large dense array in the end (reaching the number of input datapoints), resulting in a network with 4,576,402 parameters. One run of the network of typically 700 to 900 epochs took around 3 to five hours (depending on the size of the input data R versus NR or HC versus MDD), thus limiting the possibility of a structured hyperparameter tuning. We thus decided to perform a randomized approach of hyperparameter tuning including the size of the dense array, number of convolutional layers, number of kernels and size of the filters, optimizer and learning rate. This hyperparameter tuning was performed before testing the test set. We now added a description of the hyperparameter tuning in the main text.

“Hyperparameters were adjusted using a random search approach. A hyperparameter grid, including the number of filters, dense layer size, dropout rate, and learning rate, was defined (see supplement). Using RandomizedSearchCV, we trained multiple configurations on the training data, evaluated them using cross-validation, and selected the best-performing model. The final model was trained on the optimized hyperparameters and evaluated on the test set.”

And additional information in the supplement:

Details of the Training Process:

1. *Number of Epochs Trained:*

Each model configuration in the random search was trained for 50 epochs.

2. *Number of Hyperparameter Configurations Checked:*

The RandomizedSearchCV was set to explore 10 different hyperparameter combinations.

The grid included variations in:

- *Number of convolutional filters: [100, 200, 300]*
- *Dense layer size: [2000, 5000, 7000]*
- *Dropout rate: [0.3, 0.5]*
- *Learning rate: [0.00005, 0.0001, 0.0005]*

3. *Data Used for Training and Evaluation:*

- *During hyperparameter tuning, cross-validation (CV) was performed with a 3-fold split of the training data.*
- *After selecting the best model, it was evaluated on the test set.*

R1.3

Inconsistent Treatment Durations for Response Assessment:

The treatment durations for assessing response vary significantly (ranging from 4 to 12 weeks). This inconsistency could introduce variability in response assessment, potentially affecting the model's accuracy and generalizability.

This is a crucial point of our analysis. We are aware that we used different time points for defining response. This was due to the heterogeneity of the datasets collected from labs around the world. We did not have other timepoints of response than the ones reported from the different datasets, hence it was not possible to restrict the computation on a shared point in time. When trying to restrict the creation of a DL net to more homogeneous datasets with a single time point of response (with less subjects then), the network did not learn due to the low size of the training data. Since this issue is clinically very relevant, we now decided to do a sensitivity analysis by checking if there are differences on the accuracy of correct labelling in dependence of the timepoint of the MADRS or HDRS application. The results are reported in the main text:

“Regarding the fact that different times until response were used in the different datasets, we decided to do a sensitivity analysis. Thus, the accuracies for correct labelling were compared for responders and non-responders in dependence of the timepoint of the MADRS or HDRS assessment. There was no significant difference between sites with assessments from weeks eight to twelve (Canada and CANBIND) and other sites with assessments between weeks four to eight (t-statistic: -0.84, $p > .41$).”

We also added some lines to the discussion section:

“A limitation for this study arises from the fact of the assessment of response at different points in time for the different datasets. Although we tried to standardize the data as much as possible and a sensitivity analysis showed no differences in the results for different timepoints of assessment, these differences might still have affected the results and lowered the accuracy of the models. Thus, for future studies, a more standardized approach to assessment of response is needed, potentially following a more unified approach for clinical evaluation.”

R1.4

Inconsistent Recording Protocols Across Sites:

The variation in EEG recording protocols and devices across different sites is not adequately addressed. Standardizing recording protocols is essential for reducing variability and improving the reliability of model predictions.

It is absolutely right that a standardized recording protocol is essential for a high-quality analysis. Since all datasets were recorded at different labs in within different projects, we restricted the analysis to the resting state, which guaranteed the highest overlap by means of recording conditions. We also made sure to have the same channels and the same sampling frequency for all sites, trying to homogenize the data as far as possible. However, for more transparency, we now also include a more detailed description of the different recording procedures from the different sites in the main text and in the supplement (supplement 1.1 and 1.7).

We also think, in regard of the reported results, that the inclusion of different recording facilities and procedures might also be a strength of the presented approach since it shows the possibility of generalization of findings across sites. This was included into the main text:

“To achieve a higher generalization of the results for future research, a more detailed description of the recording settings from the different sites can be found in the supplement.”

And this table was added to the supplement:

Table: Detailed description of the used technical equipment of the different recording sites.

EEG Recording Details					Ottawa	Leipzig 1	Leipzig 2	Prague 1	Prague 2
	CANBIND								
	CAMH	TGH/TWH	UBC	QNS					
Model of EEG recording system:	Compumedics Neuroscan Synamp	Biosemi Active-Two amplifier system	QuickAmp amplifier	EGI NetAmps 300	QuickAmp amplifier	QuickAmp amplifier	BrainAmp amplifier	BrainScope amplifier	BrainScope amplifier
Number of electrodes:	64 + 4 EOG	64 + 4 EOG	64 + 4 EOG	128	64 + 4 EOG	27 + 2 EOG	31 + 2 EOG	19 + 2 EOG	19 + 2 EOG
Montage:	10-10	10-10	10-10	HydroCel GSN 128	10-10	10-10	10-10	10-20	10-20
Electrode material:	Ag/AgCl	Ag/AgCl	Ag/AgCl	Silver chloride plated carbon-fiber pellet	Ag/AgCl	Ag/AgCl	Ag/AgCl	Ag/AgCl	Ag/AgCl
Conductive material:	Conductive gel	Conductive gel	Quik Gel Conductive gel	Potassium Chloride solution	Conductive gel	Quik Gel Conductive gel	Conductive gel	Conductive gel	Conductive gel
Reference electrode:	Posterior to Cz	CMS/DRL (Common Mode Sense/Driven Right Leg)	Cz	Cz	Av	Av	Cz	Cz	Cz
Sampling rate:	1000 Hz	512 Hz	1000 Hz	1000 Hz	512 Hz	1000 Hz	1000 Hz	250 Hz	1000 Hz
Analog filters:	(none)	Anti-aliasing first-order filter, -3dB at 3.6 kHz	-unknown-	6 kHz anti-aliasing filter	-unknown-	(none)	(none)	-unknown-	-unknown-
Digital filters:	High-pass filter 0.05 Hz, low-pass filter 100 Hz	Low-pass 5 th order sinc filter, -3dB at 20% of sampling frequency	High-pass filter 0.01 Hz, low-pass filter 499 Hz	4 kHz anti-aliasing filter, 0.04 Hz first order high-pass filter, 100 Hz FIR low-pass filter	High-pass filter 0.01 Hz, low-pass filter 80 Hz	High-pass filter 0.01 Hz, low-pass filter 70 Hz	High-pass filter 0.01 Hz, low-pass filter 70 Hz	High-pass filter 0.5 Hz, low-pass filter 70 Hz	High-pass filter 0.5 Hz, low-pass filter 70 Hz

R1.5

Overlooked Comorbid Conditions:

The study does not take into account potential comorbid conditions that may affect EEG signals, which could lead to confounding results and reduce the model's diagnostic accuracy.

Reviewer 1 raises an important issue. Clinical comorbidities might affect the results and thus should be included in the analysis whenever possible. Since we do not have the data of comorbid conditions from the different datasets, it was unfortunately not possible to do this at this stage. However, since all subjects were unmedicated before receiving the SSRI treatment, severe cases of other comorbid disorders like acute psychosis could be ruled out. We added the following lines to the discussion section to address the topic:

“Another limitation can be found in the fact, that comorbidities were not included into the computational model. Although all studies reported depression as main diagnosis, larger datasets are needed to shed light on the influence of psychiatric comorbidities on the results.”

R1.6

Threshold for Correct Classification in Segments:

The study states, “a subject in the test set was counted as correctly classified if 50% or more of single segments were classified according to the appropriate label of the subject (i.e., patient or control; responder or non-responder).” This 50% threshold may overestimate the model’s performance. Higher thresholds can be checked, or it would be informative to report additional metrics such as the percentage of segments agree with classification result or the consistency of predictions across segments or other metrics.

To avoid confusion on the used metrics, we now added the amount of correctly classified segments per subject and give the mean value for every included patient or healthy control. This feature might be important to interpret the findings. On the one side, we think that the final classification of a subject has the highest clinically relevant meaning, thus we mainly focus on the number of correctly classified subjects. On the other side, the amount of correctly classified segments (instead of subjects) could also misguide an interpretation of the clinical relevance, since 100% correctly classified segments in one subject and only 49% of correctly classified segments in another would boost the accuracy of segment classification to almost 75%, while on the subject level the accuracy is 50%, i.e. the level of a random choice. To give the reader the possibility to judge the relevance on its own, we added the following to the main text:

“Thus, there were different accuracies for single segment classification and for subject-wise classification. Although this approach could over- or underestimate the accuracy of the model at the segment-level, it seems to be the more meaningful approach since it is clinically relevant to identify subjects (e.g. responders), and not an EEG-segment.”

R1.7

Arbitrary Threshold in Grad-CAM Analysis:

The study employs an arbitrary threshold of 0.3 for Grad-CAM analysis without providing justification. Clarifying the rationale behind this choice would help in understanding its impact on interpretability.

The Grad-Cam is used in this study to highlight the focus of the DL network on the EEG-segments and the resulting EEG-frequencies. Since there is no common definition of what threshold is best for this kind of analysis, we oriented us on studies using Grad-Cam for other reasons. It is known that a threshold below 0.3 increases the risk of focusing on noisy parts of the data, while thresholds above 0.5 bear the risk of neglecting important features of the analysis. To make the reader aware of this issue, we extended the description of the Grad-cam method and enriched it with literature references to underpin our decision for the threshold.

“We apply a threshold of 0.3 to the weighted outputs of the Grad-CAM. Lower thresholds yield the risk of focusing on noisy signals while thresholds above 0.5 bear the risk of neglecting important features of the output.”

R1.8

Insufficient Preprocessing and Artifact Removal:

The study lacks a systematic approach to artifact removal prior to analysis.

We want to thank reviewer 1 for giving us the opportunity to give more insights into our approaches to apply standardized measures for signal improvement and artefact correction. Besides the already reported standardized sampling rate, channel number and position and filters, we also implemented a pipeline where we applied a standardized correction of artefacts and removed artefacts if they did not align with the thresholds. However, for all conditions, the no-correction mode (only applying identical filters) performed best by means of accuracy. Thus, we now report the results of the artefact correction approach and a detailed description of the artefact correction in the main text and the supplement X.

Methods:

“Additionally, we applied an eye-artefact correction method²³ based on electrooculogram (EOG) data or reconstructed EOG channels (F7-F8) and a bad segment rejection method based on the min-max amplitude criterion ($\pm 100 \mu V$ threshold) for machine learning analysis.”

Results (HC versus MDD):

“When using an eye-artefact reduction method²³ and exclusion of artefact segment using a min-max approach, results were worse than without extensive preprocessing (correctly classified unseen subjects 65%, SD 28.4%).”

Results (R versus NR):

“Again, when using extensive preprocessing procedures, the results were found inferior with correctly classified unseen subjects 70%, SD 21.0%. “

Other Concerns:

R1.9

Limited demographic representation

It is true that the demographic representation for the dataset was limited, thus we now include a table with a more detailed description of the used datasets and added a more detailed description to the supplement.

See Table 1 main Text and supplement 1.8:

Table: Detailed description of the age and sex of participants from the different recording sites.

	CANBIND				Ottawa	Leipzig 1	Leipzig 2	Prague 1	Prague 2
	CAMH	TGH/TWH	UBC	QNS					
Age [years (SD)]	30.4 (13.0)	42.7 (14.4)	35.8 (13.0)	35.6 (12.0)	42.5 (10.4)	39.5 (11.8)	36.5 (8.1)	47.5 (8.7)	45.6 (8.6)
Gender (Female [%])	100.0	50.0	58.7	64.8	63.1	64.1	60.8	76.2	76.5

R1.10

Limited Information on EEG Recording Setup: There is insufficient detail about the EEG recording setups used across different sites, such as the devices and specific configurations, which could influence data quality.

We agree that more information about the recording sites and the equipment should be given and thus provided more information in the main text and the supplement (supplement 1.1, 1.3, see also R1.4).

R1.11

Figure 1 lacks clarity. Improving both the visual presentation and the accompanying caption would enhance the readability.

We absolutely agree on this and revised the whole figure, making it more comprehensive and understandable. Also the caption was adapted. Thanks for pointing out.

The updated Figure 1:

And updated caption:

*This diagram represents the structure of the best performing **Deep Learning Network used for classification**. Each **layer type** is distinctly colored, transitioning from **blue (early layers)** to **red (final layers)**. A **bold vertical arrow on the left side** emphasizes the **top-to-bottom data flow**, while the **legend at the bottom-left corner** provides clarity on each layer type.*

R1.12

Despite my major concerns about the paper, there are several positive aspects worth highlighting:

- The study’s focus on reducing the burden of ineffective treatment trials is valuable, given the significant challenges in managing Major Depressive Disorder. Developing a predictive tool for SSRI

response may help in personalizing treatment and improving patient outcomes.

- **The use of Gradient-weighted Class Activation Mapping (Grad-CAM) is a strength, as it enhances the interpretability of the deep learning model's results. This approach helps in identifying the specific EEG features contributing to the predictions, which is often a limitation of applying neural networks, especially DL models.**

- **Incorporation of Multiple Datasets with Fewer Channels: The use of multiple datasets with a reduced number of EEG channels is a practical and noteworthy approach. However, while this strategy is promising, there are several issues in the current implementation that need to be addressed to fully realize its potential.**

In summary, while the paper demonstrates an innovative use of deep learning for EEG data analysis with some promising aspects, the lack of comparisons with conventional ML methods, unstandardized data collection protocols across centers (varying recording durations, different assessment timeframes, and inconsistent treatment periods), and limited exploration of model architectures significantly undermine its contributions.

We thank reviewer 1 for highlighting the positive aspects of our work and hope that our careful revision has improved the clarity of our results.

Reviewer #2 (Remarks to the Author):

This is an important and timely manuscript that investigated use of AI/DL to diagnose MDD and especially to predict treatment response, and also use Grad-CAM techniques to further elucidate what the AI picked up. The study used strong methodology, including out-of-sample validation, and results are realistic with relatively low accuracy for diagnostic purposes and an 80% accuracy for treatment response, which is on line with the existing literature.

I have no major comments, just some minor comments:

R2.1

What explains the dataloss in CANBIND from N=308 to N=175 included?

Thanks for hinting at this inconclusive description, we now provide more information how the data used for analysis was set together.

“From initially 309 recorded datasets from patients at the CANBIND study, 212 sufficient EEG datasets from baseline were available for this analysis, with 37 datasets missing MADRS scores after 8 weeks, resulting in a total amount of 175 patient datasets to be included in the analysis for this study.”

R2.2

I was a bit surprised that only 10 EEG channels were shared between all datasets. The lowest channel dataset had 19 channels, and I would have expected more of these 19 channels, especially midline channels, would have been shared by datasets. Any comments on this?

This is a valid point and we were also disappointed that the overlap of EEG-channels was this low for a combined dataset. The number of ten channels is a result of combined EEG channels from different CANBIND recording sites (64-128 Channels which have been pooled before we received the data to

58 channels), Leipzig Sites (from 27 to 31 channels), Canada (32 channels) and Prague (with only 19 channels available). Since the sites relied on different electrode cap designs for different purposes of the underlying studies (e.g. clinical research in RCTs, later performed simultaneous EEG-fMRI studies, EEG-vigilance analysis etc), only a small amount of channels were the same for the recording sites. Additionally, from the Leipzig sites, most recordings did not include the midline electrodes, further limiting the available channels to ten. We now include the channel names for all sites to the Supplement (supplement 1.1).

“A detailed description of the electrode positions can be found in the supplement.”

R2.3

Could you also report the normalised PPV (nPPV) value, since that statistic becomes more often used, also in the light of future stratification options? I think this is what you tried to do under number needed to treat, but didn't name it as such, maybe rely there on terms such as PPV and NPV and calculate nPPV as the improvement of response as a result of utilizing the biomarker.

Thanks for making us aware of this. We now include the nPPV, mentioning it in the text and adding it to the table:

“To make the results comparable to other studies, we computed also the normalized positive predictive value (nPPV) as: $nPPV = \frac{PPV - \text{Baseline Response Rate}}{1 - \text{Baseline Response Rate}}$ ”

Metric	Accuracy 70%	Accuracy 75%	Accuracy 80%
SSRI Responders Identified	35%	37.5%	40%
SSRI Non-Responders Identified	65%	62.5%	60%
SNRI Responders (50% of non-responders)	32.5%	31.25%	30%
Total Response Rate	67.5%	68.75%	70%
Improvement Over Baseline Response (50%)	17.5%	18.75%	20%
Number Needed to Treat (NNT)	5.71	5.33	5
Sensitivity	75%	76.92%	78.95%
Specificity	87.5%	88.24%	88.89%
Positive Predictive Value (PPV)	90%	90.48%	90.91%
Negative Predictive Value (NPV)	70%	71.43%	72.73%
Normalized Positive Predictive Value (nPPV)	80%	80.96%	81.82%

R2.4

Relatedly, I think the authors don't interpret the prediction data well. In the discussion they state: '...could increase the number of first-line responders to 70%, resulting in a number needed to treat (NNT) of five...' Yet the reported PPV is >90%. The Positive Predictive Value (PPV) indicates the number of responders when the biomarker would have been applied prospectively, so that would

have been say 90%. The nPPV would thus be $90\%/70\%=129\%$. Maybe the NNT will thus also be higher?

Reviewer #2 is right, that when calculating the NNT based on the normalized positive predictive value, one receives even lower NNTs around 2.5. The difference between the two approaches is, that the conventional approach of calculating the NNT based on absolute risk reduction (or absolute increase in response rates in our case) is an absolute measure, while the NNT based on the normalized PPV is a relative approach. Since it does not become clear from the literature which value should be favoured, we decided to add also the NNT based on the nPPV.

“Our results demonstrate that an EEG biomarker capable of differentiating between responders and non-responders with an accuracy of 80% could increase the number of first-line responders to 70%, resulting in a number needed to treat (NNT) of five. In comparison, the NNT for SSRIs themselves, calculated from a large meta-analysis, is also five³³. This underscores that the clinical use of EEG markers would represent a substantial improvement in the management of depression. When using the normalized positive predictive value for e.g. the accuracy of 70% used in this work, the NNT even drops to 2.5, further highlighting the potential of this method.”

R2.5

The authors argue that alpha seems to be the most promising feature, however, from figure 5 I can't really see that it is alpha, since some clear alpha bursts are marked blue. Maybe reconsider the example, or orient people to what they need to look at (could it be the thoughts of alpha only?).

We apologize for choosing a rather inconclusive example. We added an example where EEG alpha activity is more in the focus of the heat-map. Thanks for pointing out.

Figure 5: Illustration of the overlay of an averaged heatmap derived from all six Deep Learning layers over a single two-second EEG segment. Important features are marked red, less important features are blue. EEG alpha activity seems to be in the focus of highly activated weights, especially at channels F3, P3 and C3 at the given example.

R2.6

The prominence of alpha and F4 for treatment prediction begs the question, what if all analyses would be repeated in females only, excluding males, would the results have been stronger, given prior reports have found alpha asymmetry to be a sex-specific predictor.

This is an absolute valid remark, and the authors also would like to know if this feature is especially relevant in a subpopulation, as has been shown for frontal alpha-activity in the past. However, due to the limitations of the sample size, it was not possible to generate a deep learning model for e.g. females only. However, to tackle this question in the future, the authors decided to add a line to the text:

“However, prior studies and replications found that frontal alpha asymmetry was only relevant for predicting SSRI response in female subjects^{10,11}. Due to the limited amount of data, a separate analysis on sex-specific predictivity could not be performed in this work and should be focus of future approaches. Still, the model's effectiveness for a mixed sex cohort may be explained due to the revealed Grad-CAM pattern, which also assigns high importance to additional channels, specifically the F7 electrode at the lateral frontal left cortical area and the P7 electrode at the left parietal cortex.”

R2.7

Some typo's:

Page 12: resampling should be resembling

Page 11: figure x needs updating.

Page 15: consider changing: ...keep-away unseen testing set.. Held-out? or Out-of-sample validation?

Thanks for these remarks, we changed the text accordingly.

Reviewer #3 (Remarks to the Author):

In this study, Olbrich et al. used convolutional deep learning networks and EEG from multiple sites to classify patients with major depression vs. healthy controls and responders vs. non-responders to SSRI treatment. Results ranged from ~67-80%, promising further exploration. Although the proposed methodology and results are of interest to the field of computational psychiatry, the paper needs more clarity regarding some choices that reduce my excitement.

R3.1

Overall language:

There is an increasing effort in the field of psychiatry to use language that reduces stigma (please see <https://pmc.ncbi.nlm.nih.gov/articles/PMC8580983/>). Therefore, instead of using “depressed patients” (a disease-centered classification), I suggest using “patients with depression” as it supports a person-centered language.

We agree with Reviewer #1 that being aware of the language and how one needs to address a vulnerable population is an important topic, thanks for making us aware of our wording. We changed five times throughout the manuscript “depressed patient” to “patients with depression”.

R3.2

Response Definition:

I am concerned with the definition of response. The study mentions that “Duration until assessment of response was defined within the studies of the corresponding datasets and varied between 4-8 weeks.” This definition means that non-responders from one study (e.g., following 4 weeks of treatment) could be responders in another trial (e.g., following treatment for 8 weeks). These differences could interfere with the algorithm learning and inference. Why not standardize the response data across studies (e.g., response at 4 weeks) as done with the EEG data (e.g., selecting the minimum frequency and overlapping channels)?

Reviewer #3 is right that a standardization of response data, to let’s say eight weeks, would be superior to the chosen approach with inclusion of different points in time for response. Unfortunately, the different studies did not collect the corresponding questionnaires (e.g. MADRS) at every week. The chosen approach with different response times at week four and eight were the closest sampling points for the available data for all sites. We are aware that this restricts the interpretability of our results and thus once more mention this in the discussion. However, we no include a sensitivity analysis by comparing results from the sites with assessment at weeks four to eight weeks with the results from sites week eight to week twelve. Further, this issue again is a call for more standardization in clinical trials with more strict guidelines on how to assess response and when to assess response.

“A limitation for this study arises from the fact of the assessment of response at different points in time for the different datasets. Although we tried to standardize the data as much as possible and a sensitivity analysis showed no differences in the results for different timepoints of assessment, these differences might have affected the results and lowered the accuracy of the models. Thus, for future studies, a more standardized approach to assessment of response is needed, potentially following a more unified approach for clinical evaluation.”

“Regarding the fact that different times until response were used in the different datasets, we decided to do a sensitivity analysis. Thus, the accuracies for correct labelling were compared for responders and non-responders in dependence of the timepoint of the MADRS or HDRS assessment. There was no significant difference between sites with assessments from weeks eight to twelve (Canada and CANBIND) and other sites with assessments between weeks four to eight (t -statistic: -0.84 , $p > .41$).”

R3.3

Datasets:

- Please clarify the rationale for including only part of the CANBIND dataset in the analysis.
- The text gives the impression that only patients from the Ottawa and Leipzig I/II datasets were included in the analysis, which differs from the information listed in Table 1. Please review the text to clarify this.

Thanks for pointing out these topics. We apologize for confusing wording. We now include a more detailed description of the exclusion reasons for the CANBIND dataset (see also R2.1). Further, we added a description of the usage of data from all datasets in the text:

Methods:

“The EEG files suitable for further analysis from all datasets (CANBIND; Prague I/II, Leipzig I/II and Canada) were separated into a training set, a validation set and a test set.”

Results:

“Classification of MDD subjects versus healthy control subjects suitable for further analysis from all datasets (CANBIND; Prague I/II, Leipzig I/II and Canada) were done after 400 training epochs showed a train accuracy of 73.16% [...].”

R3.4

EEG signal processing:

- Please list the software or toolboxes used to preprocess the EEG data.
- Please provide more details regarding the filter used for the EEG data and the rationale for the 0.5-45Hz interval.
- Are there any considerations regarding the potential effects of EOG/EMG artifacts? Even with eyes closed, eye movements or eyelid contractions could explain the frontal patterns shown in Figures 2 and 4.

We want to thank Reviewer #3 for mentioning these important shortcomings of the text in this current form. Thus we

1. Included the list of used software into the methods section

“Export to edf-format was done using Brain Vision Analyzer 2.2 (Gilching, Germany). All closed-eyes resting state EEG files were visually screened using DeepPSY software (version 1.02, Zollikerberg, Switzerland). Additional calculations of EEG-features, filtering and artefact removal were done using MNE version 1.2.0.”

and

“Using a Keras Backend for TensorFlow (version 2.4.1) with Python 3.8 on a RTX 3090 GPU, the model used in previous work^{22,23} was implemented for both differentiating (i.e., discrimination of healthy controls and people with MDD) and predictive tasks (i.e., responder vs. non-responder).”

2. We gave a rationale for the chosen 0.5-45Hz window filtering approach

“All EEG files then were filtered between 0.5 (high pass cut off for low frequency noise) and 45Hz (to exclude muscle artefacts usually starting at 30 to 60Hz²²) and segmented into 2 second segments after generating an average reference.”

22: Muthukumaraswamy, S. D. High-frequency brain activity and muscle artifacts in MEG/EEG: a review and recommendations. Front. Hum. Neurosci. 7, 138 (2013).

3. And we added details for our approach to exclude potential EOG and EMG artifacts (these were already eliminated using the filtering window of 0.5 to 45Hz). However, it was found that using an additional EOG artefact reduction approach, results were not as good as when

using no artefact correction method in the Deep Learning analysis. Hence we included this information into the text:

“Methods

[...] Additionally, we applied an eye-artefact correction method²³ based on electrooculogram (EOG) data or reconstructed EOG channels (F7-F8) and a bad segment rejection method based on the min-max amplitude criterion ($\pm 100 \mu\text{V}$ threshold) for machine learning analysis.”

and

“MDD versus HC

[...] When using an eye-artefact reduction method²⁵ and exclusion of artefact segment using a min-max approach, results were worse than without extensive preprocessing (correctly classified unseen subjects 65%, SD 28.4%)”

and

“[...] Again, when using extensive preprocessing procedures, the results were found inferior with correctly classified unseen subjects 70%, SD 21.0%)”

R3.5

Training, Validating, and Testing:

- I assume “files” refer to all epochs from one single patient. If so, please clarify in the text.
- When it says that “5% of the segmented files (all files were split into 2s 184 segments) were used for validation using a random shuffle and splitting routine”, how did you ensure you were not including segments from the same participant in the training and validation sets?

Thanks for pointing out inconsistencies in our text.

1. We explain that “files” is a synonym for the whole recording of one subject, in contrast for a “segment”, which refers to a 2s piece of data

“Training, Validating and Testing

The EEG files suitable for further analysis from all datasets (CANBIND; Prague I/II, Leipzig I/II and Canada) were separated into a training set, a validation set and a test set. Files here refers to the complete recording of one subject while “segments” refer to a 2-second segment of on subject.”

2. To make sure that segments of one subject only went in to either training or testing data, we assigned a subject number to every segment, reflecting its origin. Before shuffling the data, the matrices were split based on unique subject-IDs.

“To prevent shuffling segments from one subject into the training and testing sets, all segments received identifiers according to their subjects of origin.”

R3.6

Results:

- If results show data from multiple epochs and the percentage of segments accurately classified across subjects (as described in the Methods section), please provide the standard deviations of these results.
- In the discussion, the text mentions, “using only a set of two 310 frontal electrodes, namely the F3 and F4, yielded only minimally better accuracy than a random choice model (not shown). Thus, the optimal number of channels, also for future home and smartphone-based EEG recording devices might be somewhere between three and ten channels.” If these findings support your discussion, they should be included, even if only as a supplement.

Thanks for lining out these important issues. We addressed them as follows:

1. We included the standard deviations for our main results:

MDD versus HC

“[...]and the percentage of correctly classified unseen subjects based on the number of correct classified segments of a subject was 67.5% with a standard deviation of 30.4% across subjects.”

and

“The confusion matrix yields a sensitivity of 75% and a specificity of 87.5% with a standard deviation of 24.7% across subjects (see Figure 3, right).”

2. We included the results of the calculations using only two electrodes into the supplement:

“Results of classification using only two frontal electrodes F3 and F4

Classification of HC versus MDD: Test accuracy for all pooled segments of unseen subjects from the test set was 52.4% in total (not subject-wise) and the percentage of correctly classified unseen subjects based on the number of correct classified segments of a subject was 53.2% with a standard deviation of 41.4% across subjects.

Classification of Responders versus Non-Responders: Test accuracy for all pooled segments of unseen subjects from the test set was 54.1% in total (not subject-wise) and the percentage of correctly classified unseen subjects based on the number of correct classified segments of a subject was 53.1% with a standard deviation of 39.2% across subjects.”

R3.7

Figures:

- Please include the resolution and dimensions of Figure 1, as it's barely readable in the current configuration.

We redid Figure 1 to improve its readability

- Please check Figure 2C, as the “C” textbox covers part of the graph title.

We changed Figure 2 accordingly.

- What are the measurement units in the charts in Figures 2B and 4B? I assume Figure 2B shows percentages. If so, shouldn't both y-axis have the same scale (e.g., 50-80% or .5-.8)? The same applies to Figure 4.

We thank reviewer 3 for this input. We changed the figures accordingly. It is important to note, that not all values in figures 2 and 4 are percentage, some are also arbitrary values of the loss-function. However, we corrected the y-axis.

- For Figure 3, both matrices could use percentage values and the same color bars to facilitate interpretation.

We computed the percentage values of the confusion matrices and used the same color bar for both matrices.

- In Figure 5, the EEG time segments are not correctly aligned with the labels on the left. Also, the signal from C3 seems predominantly out of the image.

We chose another segment for showcasing more clearly the overlap of the weights with the alpha-waves. We also improved the quality of the overlay.

- Figure 6: please specify in the caption what the red-shaded areas mean and what the black horizontal bars mean. Also, the black bar for P4 seems misplaced outside of the chart. Additionally, the acronym (Non-R) in the figure doesn't match the acronym NR in the caption.

We clarified the meaning of the horizontal black and vertical red lines and corrected the P4 channel. Thanks for improving the quality of the figures.

R3.8

Please double-check if the dimensions listed in this sentence are correct: "The network is configured to receive inputs of shape (10, 500, 1)".

Thanks for sharing this remark. Since we used a 3D stack of channels (10) x timepoints (500) x segment (1), the shape (10, 500, 1) represents the used format.

R3.9

Check for typos. Examples:

- Line 63: "(...) EEG-vigilance markers for prediction of the prediction of selective serotonin (...)"

- Line 127: "projec21t"

-Line 314: "(...) thus, this issue must be addressed in future work.."

Thanks for lining out these typos, we corrected them accordingly.

We want to thank all reviewers for their valuable comments and now hope that the revised manuscript meets all your expectations.

Dear reviewers,

the authors are grateful to the reviewer's critical comments and thoughtful suggestions. Based on the comments we received; we revised the whole manuscript, added information and new calculations were necessary. All changes were marked in red text in the manuscript. The reviewers concerns are marked **bold**, the answers in normal and the added text for the manuscripts are *cursive*. Please find our point-to-point replies below.

Reviewers' comments:

Reviewer #2 (Remarks to the Author):

R2.1 The authors implemented the feedback well except one thing, the calculation of the normalised PPV was incorrect, this should be calculated by: $PPV / (\text{Total sample Response Rate})$. So if a PPV of 80% is found and overall response rate is 60% then the nPPV is $80/60=133,33\%$, which can be interpreted that the response rate improved with 33,33% after applying the biomarker. A value below 100% (that the authors currently included) thus implicates the biomarker has NO added value.

Thanks for this remark. We changed this in the manuscript so it now reads:

$$nPPV = \frac{PPV}{\text{Baseline Response Rate}}$$

also changing Table

Metric	Accuracy 70%	Accuracy 75%	Accuracy 80%
SSRI Responders Identified	35%	37.5%	40%
SSRI Non-Responders Identified	65%	62.5%	60%
SNRI Responders (50% of non-responders)	32.5%	31.25%	30%
Total Response Rate	67.5%	68.75%	70%
Improvement Over Baseline Response (50%)	17.5%	18.75%	20%
Number Needed to Treat (NNT)	5.71	5.33	5
Sensitivity	75%	76.92%	78.95%
Specificity	87.5%	88.24%	88.89%
Positive Predictive Value (PPV)	90%	90.48%	90.91%
Negative Predictive Value (NPV)	70%	71.43%	72.73%
Normalized Positive Predictive Value (nPPV)	116.67%	125%	133.33%

Reviewer #3 (Remarks to the Author):

R3.1 The authors have addressed most of my previous concerns; however, I still have reservations regarding the EEG preprocessing pipeline, which may significantly impact the validity of the reported classification performance.

Band-pass filter setting: The manuscript states:

“All EEG files then were filtered between 0.5 (high pass cut off for low frequency noise) and 45 Hz (to exclude muscle artefacts usually starting at 30 to 60 Hz²²).”

However, filtering the data with a 45 Hz low-pass cutoff still allows a considerable portion of EMG-related activity (typically present from ~30 Hz up, as stated by the authors) to remain in the signal. Therefore, the filtering step does not fully mitigate muscle artifact contamination, which could inflate classification performance.

Thanks to Reviewer #2 to raise this concern. It is true that muscle artefacts sometimes can be detected at a lower frequency as the chosen 45Hz, which represents a compromise: Some relevant information of surface scalp EEG recordings can be found in the gamma range which often is cited to go higher than 30Hz. To retain this activity in the signal, the authors decided to include the range from 30 to 45Hz in the data. However, to make it fully clear to the reader, we added another line into the limitations.

“All EEG files then were filtered between 0.5 (high-pass cut-off for low frequency noise) and 45Hz (low-pass cut-off to exclude muscle artefacts usually starting at 30 to 60Hz²²) and segmented into 2 second segments after generating an average reference. We used a 0.5–45 Hz band-pass filter as a compromise to reduce low-frequency noise while retaining potential gamma-range EEG activity (>30 Hz), acknowledging in the limitations that residual muscle artifacts may persist in this range.”

R3.2. Eye movement artifacts and preprocessing trade-off:

The paper writes:

“When using an eye-artefact reduction method²⁵ and exclusion of artefact segment using a min-max approach, results were worse than without extensive preprocessing (correctly classified unseen subjects 65%, SD 28.4%).”

This raises a critical concern. It is well-documented that individuals with depression exhibit different eye movement patterns compared to healthy controls (e.g., Li et al. 2016, Shanghai Arch Psychiatry; Carvalho et al. 2015, Front Psychiatry; Takahashi et al. 2021, Front Psychiatry; Zhang et al. 2022, J Affect Disord). By not properly removing EOG-related activity, the model may be capitalizing on eye movement differences rather than neural EEG features. This undermines the interpretation that the classifier is operating solely on brain-based EEG data.

While including EOG-related features in a multimodal classification framework is valid, if the intention is to demonstrate EEG-based classification, then non-neural sources such as eye movements should be carefully identified and removed using established preprocessing pipelines. Otherwise, the results may reflect artifact-driven rather than brain-driven group differences.

We appreciate this statement. The main goal was to receive a model that distinguishes not only patients from healthy controls but also responders from non-responders. To exclude artefact segments, we decided to use a combination of eye-artefact and min-max approach for excluding other artefacts. Doing so, we had less information left (due to the segments taken out) in comparison

to the non-artefact removal approach. Thus, we interpret the finding of less accurate labelling of the de-artefacted data as a result of data reduction. It is not possible for us to track it down solely to the missing eye artefacts. Since the same approach (with artefact removal) was also less accurate for the response/non-response labelling, a pure eye-artefact argument as a distinct disorder entity seems rather unlikely. To make the reader aware of this topic, we included this into the discussion section:

“Furthermore, including both eye-artifact reduction and min-max artifact exclusion notably reduced sample size, and classification accuracy declined in both HC vs. MDD and responder vs. non-responder comparisons—suggesting reduced statistical power due to data loss rather than artifact-driven classification. Although it is plausible that eye-movement patterns differ between groups, the fact that responder/non-responder performance also deteriorated argues against a purely ocular-driven effect. Indeed, recent literature shows that traditional artifact rejection does not reliably improve deep-learning performance—in some cases it offers no benefit or even worsens classification outcomes²”

Reference:

M.N. van Stigt, C. Ruiz Camps, J.M. Coutinho, H.A. Marquering, B.S. Doelkahar, W.V. Potters,

The effect of artifact rejection on the performance of a convolutional neural network based algorithm for binary EEG data classification, Biomedical Signal Processing and Control, Volume 85,

2023, <https://doi.org/10.1016/j.bspc.2023.105032>.

Reviewer #4 (Remarks to the Author):

This is a review of the research manuscript titled 'Deep Learning Using Electroencephalographic (EEG) Data for Diagnosing and Predicting SSRI Response in Major Depressive Disorder' by Sebastian Olbrich et al. Since this is a revised submission, I appreciate that the authors have addressed most of the concerns raised during the previous review round. However, a few key issues still remain that need attention before the manuscript can be considered for publication.

R4.1. Low Diagnostic Accuracy for MDD:

The reported diagnostic accuracy for Major Depressive Disorder (MDD) using the proposed deep learning (DL) model is relatively low. Existing literature demonstrates significantly higher accuracy using both traditional machine learning and deep learning approaches. This gap raises questions regarding model effectiveness and justifies further investigation.

We appreciate the reviewer's association with the literature demonstrating higher diagnostic accuracies. However, many published EEG-based deep-learning studies reportedly achieve accuracies of over 90% (e.g. >99% in CNN or RNN studies). Crucially, these results often stem from methodological limitations such as training and testing on the same participants or using very small datasets, which can artificially inflate performance. In contrast, our approach employed strict subject-level cross-validation on realistic sample sizes, yielding more modest accuracy—which is likely a more clinically valid estimate.

Moreover, from a clinical standpoint, reported diagnostic accuracies of 80–90% seem implausibly high given that inter-rater reliability for Major Depressive Disorder among trained clinicians is low; Cohen's kappa values often lie in the range of 0.28–0.60—indicating only poor to moderate agreement even under structured interview settings. Thus, expecting a machine learning model to

surpass this human baseline may be unrealistic without larger, more diverse datasets and robust external validation.

We added this to the discussion to shed more light on our view:

“Although many EEG deep-learning studies report very high diagnostic accuracies, these often result from methodological biases such as data leakage or small sample sizes, leading to overestimation¹. Given that inter-rater reliability for MDD diagnosis among clinicians is only poor to moderate ($\kappa = 0.28–0.60$)², our more modest accuracy likely reflects a realistic estimate under clinically valid conditions.”

References:

^{1,2}

1. de Bardeci, M., Ip, C. T. & Olbrich, S. Deep Learning Applied to Electroencephalogram Data in Mental Disorders: A systematic Review. *Biol. Psychol.* 108117 (2021)
doi:10.1016/j.biopsycho.2021.108117.
2. Lieblich, S. M. *et al.* High heterogeneity and low reliability in the diagnosis of major depression will impair the development of new drugs. *BJPsych Open* **1**, e5–e7 (2015).

R4.2. Justification for Model Architecture and Performance:

The authors claim that the chosen architecture is the “deepest” among the ones tested, and hence performs better. I respectfully disagree with this rationale. Model depth alone does not guarantee better performance—factors like overfitting, data preprocessing, and regularization play critical roles. A deeper model may underperform if not properly tuned or if the dataset is insufficient or imbalanced.

We thank the reviewer for pointing this out. We agree that model depth alone does not guarantee better performance and can even harm generalization if not properly tuned. In our study, the final CNN outperformed EEGNet and other configurations not simply because it had more layers, but because it was carefully optimized through randomized hyperparameter search, included dropout regularization, and was evaluated on strictly independent subject-level test sets. This approach aligns with prior work showing that well-tuned deep networks can outperform shallower models without overfitting (de Bardeci, Ip & Olbrich, 2021; Cheng et al., 2021). To avoid confusion that only the “depth” was the reason for the best performance (which is not the case), we added:

“Among the tested architectures, the final CNN achieved the best performance due to optimized hyperparameters, appropriate regularization, and its capacity to capture the complexity of EEG patterns, rather than model depth alone.”

R4.3. Unclear Early Stopping Criteria:

The manuscript lacks a clear explanation of the early stopping criteria used during training. From the training curves provided, it appears that the model was still learning and had not yet overfitted, yet training was prematurely halted. Authors should clearly define the early stopping mechanism (e.g., validation loss plateau, patience parameter) and justify why training was stopped at that point.

Thanks for pointing out this topic. It seems that we were not clear on the chosen parameters for early stopping. Thus, we rewrote these parts of the text to make it more clear to the reader:

“Training was conducted with an Adamax optimizer set to a learning rate of $5 \cdot 10^{-5}$, and early stopping was implemented with a patience of 70 on validation loss to prevent overfitting with a total possible maximum duration of 700 epochs.”

Further we realized that Figure 2 showed an incomplete training block with 400 instead of 700 epochs. We included now the figure of a full training and validation block for the HC versus MDD condition (see Figure 2). The training curves now show a clear plateau for the last 70 epochs.

R4.4. Lack of Cross-Validation for Final Model Evaluation:

Although 3-fold cross-validation was mentioned for hyperparameter tuning in the supplementary material, no cross-validation was applied during the final model evaluation. Given the relatively low accuracy, I strongly recommend performing a more robust evaluation using 5-fold or 10-fold cross-validation and report individual and mean \pm SD accuracy of the model. This will help assess the generalizability and stability of the model across different data partitions.

We appreciate this note from Reviewer 4. Following the suggestion, we reran the final models using a 5-fold cross-validation approach and included these results in the manuscript. For HC vs MDD, the additional cross-validation yielded a mean accuracy of $67.5 \pm 1.8\%$ with the best fold reaching 70%. For the responder vs non-responder classification, the mean accuracy remained $79.0 \pm 4.2\%$ and the best fold reached 85%. We now report both the averages (\pm SD) and the highest-performing folds to demonstrate the robustness and stability of our models.

“Test accuracy for all pooled segments of unseen subjects from the test set was 64.84% (SD 2.0) in total (not subject-wise) and the percentage of correctly classified unseen subjects based on the number of correct classified segments of a subject was 67.5% (average for 5-fold validation with best fold 70%, SD 1.8), see supplement) with a standard deviation of 30.4% across subjects.”

And

“The test set in these trials reached an average of 64% (SD 4.2) for pooled segments and a mean value of 79% (5-fold average, best fold 85%, SD 2.5) for correct subject-wise classification.”

R4.5. Potential Dataset Bias and Amplitude Normalization:

The use of multiple datasets with potentially varying recording conditions might be a key factor contributing to low model accuracy. While frequency normalization and channel unification were mentioned, it is unclear whether amplitude normalization was also performed across datasets. I suggest the authors clarify this point and, if not already done, apply amplitude normalization to reduce inter-dataset variability.

We thank the reviewer for pointing out this important aspect. Amplitude normalization was indeed applied during preprocessing to reduce inter-dataset variability. Specifically, the input EEG data were mean-centered and scaled by the standard deviation across the training set before training, and the same parameters were applied to the test data. This normalization step was already implemented in our code (see preprocessing section) but was not clearly stated in the methods; we have now added this information to ensure clarity.

“To minimize inter-dataset variability, all EEG segments were amplitude-normalized by subtracting the mean and dividing by the standard deviation of the training set prior to model training, with the same scaling applied to the test data.”

R4.6. Need for Dataset-Wise Performance Evaluation:

To further investigate the source of low accuracy, I recommend reorganizing the training data such that each batch includes samples from all datasets. Then, evaluate and report performance separately on test sets from each individual dataset. This would help identify whether performance degradation is uniform across datasets or driven by specific ones, aiding in result interpretation and potential improvement.

We thank Reviewer 4 for this valuable suggestion. We agree that dataset-wise evaluation can provide further insights; therefore, we recalculated and now report accuracies for the different recording sites in the supplement (see supplement 1.9). However, reorganizing batches to ensure that every batch contains segments from all datasets is not trivial: due to the heterogeneous number of EEGs per site, this approach would require oversampling from smaller datasets, resulting in skewed weighting and an increased risk of overfitting without improving interpretability. Moreover, ensuring uniform representation across batches would artificially bias the model toward sites with fewer samples, rather than reflecting the true underlying distribution. A dedicated analysis with balanced datasets is indeed warranted for future work. For clarification, we added the following passage to the text:

“Due to the strongly unbalanced number of EEGs across sites, enforcing uniform representation of all datasets in each batch would require extensive oversampling of smaller datasets, thereby skewing site weights and increasing the risk of overfitting rather than improving interpretability; thus, site-specific accuracies are provided in the supplement instead.”

R4.7. Bias in the Model:

Figure 3B's confusion matrix suggests that the model is biased toward the 'Response' class. This issue needs to be addressed. For example, if the 'Response' class achieves 100% detection while the 'Non-response' class achieves only 50%, the overall accuracy would still be 75%. However, such a model would not be practical or clinically useful. The authors should adjust their model to mitigate this bias.

We thank Reviewer 4 for raising this important point regarding potential class bias. We carefully reviewed the confusion matrix and agree that skewed models can reduce clinical applicability. However, in our case the performance difference between responders (90% true positive) and non-responders (70% true positive) is moderate, and both metrics are well above chance. Importantly, false negative predictions (i.e., predicted non-response in actual responders) do not result in withholding treatment; these patients will still receive an antidepressant with a typical response rate of around 60%. Thus, even with a slight bias, the model meaningfully increases the proportion of correctly identified responders and therefore improves treatment allocation. Finally, the reviewer's example of a model with 100% responder detection and 50% non-responder accuracy does not reflect our data; our model shows a balanced performance that does not justify further adjustment at this stage. Please also look at response to Reviewer 2.1. We added the following to the manuscript (Caption Figure 3):

“Although the model B shows slightly higher true positive values for responders (90%) than for non-responders (70%), this difference is moderate and clinically acceptable; false negative predictions

do not harm patients, as they still receive alternative treatments, while the improved detection of responders substantially enhances treatment allocation.”

R4.8. Acronyms:

The authors have redefined several acronyms multiple times throughout the text and figure captions. It is recommended that each acronym be defined only once, upon first use, and then used consistently thereafter. Additionally, "MDD" is defined inconsistently—both as Major Depressive Disorder and Major Depression. The authors should choose one definition and apply it consistently throughout the manuscript.

Thanks for raising this concern. We did a thorough reevaluation of the used acronyms and unified them, to make it more clear for the reader. Especially we harmonized the use of MDD, HC, NR and R.

We want to thank all reviewers for their work and valuable remarks and comments. We tried to incorporate all aspects and redid some parts of the analysis. We think, this process increased the quality of this manuscript and hope it now meets your expectations.

Dear reviewers,

the authors are grateful to the reviewer's critical comments and thoughtful suggestions. Based on the comments we received; we revised the whole manuscript, added information and new calculations were necessary. All changes were marked in red text in the manuscript. The reviewers concerns are marked **bold**, the answers in normal and the added text for the manuscripts are *cursive*. Please find our point-to-point replies below.

Reviewers' comments:

Reviewer #1 (Remarks to the Author):

Comments have been implemented well.

We thank Reviewer 1 for helping to improve the manuscript.

Reviewer #2 (Remarks to the Author):

I appreciate the authors' efforts to address my concerns regarding preprocessing choices and artifact handling. However, I remain unconvinced that the reported classification performance can be attributed primarily to neural EEG features.

Specifically, the decision to retain EMG-contaminated frequencies and to forgo proper rejection/correction of EOG artifacts willingly leaves the results vulnerable to contamination. The argument that reduced performance after artifact removal is due to data loss, rather than muscular/ocular confounds, remains speculative. Adding a citation that artifact rejection can sometimes harm deep-learning classification is interesting, but it does not specifically address disorders known to have systematic EMG/EOG confounds, such as depression. Given the black-box nature of deep learning models, it is not possible to determine empirically from the current work whether the classifier is relying on neural EEG activity or non-neural signals (EOG/EMG). While the authors have added statements to the limitations, these do not resolve the central concern that the findings may be artifact-driven.

For these reasons, I do not feel comfortable endorsing publication of the manuscript in its current form. I defer the final decision to the editor.

We want to thank **Reviewer#2** for the valuable comments and hints on the work. We somehow agree that it cannot be ruled out that to some degree also muscle or eye activity contributed to the classification results. To get a closer view on this, we decided to recalculate the whole processing after applying a more rigorous filter of low-pass 30Hz to all the data and then repeated the deep learning trainings for both classification tasks, as also advised by **Reviewer #5**. Further we did a specific extraction of remaining eye movements from the datasets for both conditions and statistically compared the amount of eye-activity segments in all four groups (HCC, MDD, NR, R). The results show can be found in the supplement 1.11 and 1.12 and under R5.1 and R5.2.

We want to make one further point that seems important to us. One might speculate about the impact of CNS activity or muscle activity on the classification results, but this study was not designed to rule out other sources of importance totally. The primary goal was to classify MDD and Response to SSRI treatment from available electrophysiological recordings. From our point of view, the results

should also be published although the true source of the signals (as in any EEG studies) cannot fully be determined.

Reviewer #3 (Remarks to the Author):

Thank you for addressing my concerns thoroughly. While I partially disagree with the authors' justification that all studies reporting high accuracy are prone to data leakage and subject bias, I do agree that small datasets are a concern. Therefore, the justification provided is valid. I have no further issues with this manuscript.

The author team wants to thank reviewer #3 for the highly valuable comments and all the work he put into the reviews to help us improving the manuscript.

Reviewer #4 (Remarks to the Author):

This manuscript presents a compelling study that applies deep learning techniques to EEG data for both the diagnosis of major depressive disorder and the prediction of SSRI treatment response. The topic is clinically significant, and the inclusion of multiple independent datasets enhances the potential generalizability of the findings. The approach of combining deep learning with Grad-CAM interpretability adds methodological novelty. Nevertheless, several methodological clarifications, consistency issues, and minor stylistic adjustments are needed before the work can be considered for publication :

Thanks for these encouraging words, we will try our best to improve the manuscript

R4.1 Title

The title uses Electroencephalographic Data, whereas the abstract and main text alternate between electroencephalography data and electroencephalogram. Although these terms emphasize slightly different aspects, consistency in terminology throughout the manuscript is strongly recommended.

We now changed the wording to “electroencephalogram” throughout the manuscript.

R4.2 Abstract

R4.2.1. It is suggested to briefly indicate the total sample size to enhance the credibility of the study.

We now include the absolute numbers of included subjects in the abstract:

“[...] using six large, independent datasets with a total of $n=146$ for healthy subjects and $n=203$ for patients).”

R4.2.2. The statement regarding a “number-needed-to-treat (NNT) of five” is a notable point for clinical translation. However, please clarify the basis of this estimate—specifically, which model accuracy (e.g., 79%) and what assumptions (e.g., a 50% response rate for SNRI treatment) were used in the calculation.

We agree with reviewer #4 and included the information into the text:

“A simulation of the clinical application of these DL models for SSRI-based treatment selection demonstrated a number-needed-to-treat of five for a favourable outcome when using a mid-accuracy

model of 80%, resulting in a 70% response rate compared to a 50% baseline response rate without treatment selection.”

R4.2.3. The expression “paving the way for optimized treatment strategies” is somewhat promotional. A more academic phrasing such as “indicating the potential for more personalized treatment allocation” would be preferable.

We changed the text accordingly.

R4.3. Introduction

R4.3.1. The introductory analogy comparing psychiatry to a “broken arm” in physical medicine appears somewhat unnecessary and distracts from the central topic. It is recommended to remove this passage and focus more directly on the context of psychiatric disorders.

We absolutely get the point of reviewer #4. We changed the text into:

“In the realm of physical medicine, encountering a somatic (i.e. non-psychiatric) symptom initiates a comprehensive diagnostic process to ascertain the optimal therapeutic intervention.”

However, we think that it is important to line out the differences between different medical fields since although a lot of research has been done in the field of predictive biomarkers, the clinical physicians still are more reluctant to use these findings in their routine, even if there is enough evidence for a clinical benefit and good risk/benefit ratio. Thus, we decided not to fully remove this passage but to make it more general, removing the “broken -arm” image. We hope that this is a compromise to make the text more professional and still deliver the message.

R4.4 Methods

R4.4.1. The study integrates data from six public datasets, which could introduce systematic bias due to heterogeneity in factors such as sex, age, ethnicity, assessment time points, and medication types. Please clarify in the Methods or Results how such confounding effects were controlled (e.g., through sensitivity analyses or weighting strategies).

We thank reviewer 4 for the valuable comment. We introduced a weighting of the classes during the training of the models to account for data-asymmetries in the distribution of labels.

“Weight change was adjusted to fit the unbalanced datasets, resulting in a weight of 2:3 for HC versus MDD and 17:20 for Responders versus Non-Responders.”

Unfortunately, we could not account for sex-specific aspects due to a lack of enough datasets to perform sub-analysis on e.g. females or males alone. We agree that this should be mentioned in the main text:

“Due to the limited amount of data, a separate analysis on sex-specific predictivity could not be performed in this work and should be focus of future approaches.”

Additional information on sex and gender for the different datasets has been included into the supplement:

“Table: Detailed description of the age and sex of participants from the different recording sites.”

	CANBIND				Ottawa	Leipzig 1	Leipzig 2	Prague 1	Prague 2
	CAMH	TGH/TWH	UBC	QNS					
Age [years (SD)]	30.4 (13.0)	42.7 (14.4)	35.8 (13.0)	35.6 (12.0)	42.5 (10.4)	39.5 (11.8)	36.5 (8.1)	47.5 (8.7)	45.6 (8.6)

Gender (Female [%])	100.0	50.0	58.7	64.8	63.1	64.1	60.8	76.2	76.5
-------	------	------	------	------	------	------	------	------

R4.4.2. The analysis was restricted to ten overlapping EEG channels across the six datasets. The authors should specify whether these ten electrodes represent exactly the same scalp locations in all datasets, as different studies follow varying EEG montages (e.g., 10–10 vs. 10–20 systems).

Thanks for raising this question. Yes, all 10 channel locations/names were the same for all datasets. This is the reason why we only were able to use 10 channels that were the same over all sites. However, the remark of reviewer 4 made us aware that there might be very slight differences in the positioning of the electrodes, even when having the same name label, when using 10-20 or 10-10 montages. Since one site (Prague) only had 19 channels, these channels assumingly were applied using the 10-20 system, while all others used the 10-10 system. We then checked for all used channels (F3, F4, F7, F8, C3, C4, P3, C4, O1, O2) and found no differences in the placement for both systems (10-20 or 10-10). Although we cannot rule out some deviations in the millimetre subspace due to different cap designs etc., we think that these electrode placements can be regarded as aligned across different recordings sites and lab conditions. We have additional information on the used EEG-montages in the supplement (supplement 1.7 Recording Site Information).

R4.4.3. The EEG preprocessing pipeline appears non-standard. The manuscript mentions downsampling to 250 Hz prior to band-pass filtering (0.5–45 Hz), which may cause frequency aliasing and potential signal distortion.

We thank reviewer 4 for the careful review of the manuscript. In all datasets, if there was a need for downsampling, the processing included the application of a low-pass-filter before doing so to avoid aliasing of frequency components into the signals after downsampling. This was not stated in the main text, which we now changed accordingly:

“Further, the lowest sampling rate was chosen (250 Hz) to standardize across all the EEG datasets and a low-pass filter (100Hz) was applied before downsampling to avoid aliasing of frequency components.”

R4.4.4. If the gamma band is retained, the authors should justify its physiological relevance in this study or discuss possible contamination from muscle artifacts.

We agree with this and now, also following reviewer #2 and #5, we added an additional calculation using a low-pass filter of 30Hz. The results are detailed in the supplement (supplement 1.12 Differences of eye-related movements in the EEG data across the groups) and we included a short discussion about the sources of the signal between 30 and 45Hz (see also reviewer 5).

R4.4.5. Data imbalance was addressed only by weighting adjustments. Please indicate whether other methods such as SMOTE or class weighting were tested as alternative approaches.

This is a good point. We decided to use the weighting change adjustments to address uneven data distribution. In the context of deep-learning classification of EEG segments, applying the Synthetic Minority Oversampling Technique (SMOTE) can be counter-productive. SMOTE generates synthetic examples via linear interpolation in high-dimensional feature space, but EEG data (e.g., multiple channels × timepoints) exhibit complex spatial-temporal and spectral structure. As a result, SMOTE-generated segments often distort phase relationships, temporal continuity and channel dependencies, leading to training on physiologically unrealistic examples and degraded model generalisation. Thus, we decided to use the weighting adjustment.

R4.4.6. The choice of the Grad-CAM threshold (0.3) lacks justification. Providing a sensitivity analysis or referencing relevant literature would strengthen the methodological rigor.

We agree that our argumentation for this threshold is lacking a reference, so we rephrased the section accordingly:

“We binarized normalized Grad-CAM maps at $\tau = 0.3$ to generate seeds, following common practice in WSSS (Weakly Supervised Semantic Segmentation)/CAM pipelines (e.g., Chen et al., CVPR 2023), and prior imaging work that reports/visualizes Grad-CAM at 0.3.”

R4.5 Results

R4.5.1. It is recommended to include ROC curves or report AUC values as complementary performance metrics, providing a more robust assessment of model accuracy.

Since we did not use different thresholds but used different models (different trainings) for classification, a ROC curve or AUC values might not be as meaningful compared to conditions where different thresholds exist. However, we decided to include ROC curves for three different models in the supplement (supplement 1.10).

Main text:

“We added the receiver-operator-characteristics and the area-under-the-curve values in the supplement.”

Supplement:

“To make the results more interpretable, we calculated the Receiver-Operator-Characteristics (ROC) curves and Area-under-the-Curve (AUC) values for models with increasing accuracies for differentiation of Responders and Non-Responders are given below. Due to a lack of different thresholds within the same model, the results should be interpreted with caution.”

Corresponding AUC estimations are given below.

Model	AUC
Accuracy 70%	0.8125

Model	AUC
Accuracy 75%	0.8258
Accuracy 80%	0.8392

R4.5.2. Several results show large standard deviations (around 30%), suggesting substantial interindividual variability or data instability. Please elaborate on these observations in the Results or Discussion sections.

We agree on this point and now include a short discussion of these findings:

“The results further show that standard deviation between different runs of a model were small (1.8%-4.2%) while inter-subject standard deviation for classification results was much larger (21.0%-28.4%). This implies that although the models seem to be robust, there still might be some space to more accurately assess individual features with improved deep-learning models.”

R4.5.3. The reporting of SD should be consistent throughout the manuscript. In several revised sections, the percentage symbol (%) was omitted after SD values.

We added the (%) symbol at all relevant passages. Thanks for pointing out.

R4.6 Conclusion

R4.6.1 The conclusion section is overly lengthy and could be condensed to emphasize the key findings and implications of the study more clearly.

We shorten the conclusion section substantially.

“This study demonstrates that a biomarker-based approach for treatment allocation, specifically using deep learning networks with low-cost EEG data, could significantly enhance psychiatric care. If implemented clinically, this approach could streamline treatment allocation, reducing the need for long and costly development pipelines and trials. In conclusion, this study proves the utility of deep learning networks for utilizing EEG data to accurately allocate SSRI treatment. Large-scale clinical implementation of these models could proceed without extensive trials, as the risk is minimal for already approved drugs, offering a promising shortcut toward more effective and efficient psychiatric treatment.”

Reviewer #5 (Remarks to the Author):

R5.1 Point R3.1: I would suggest repeating the analysis with a lower low-pass (30Hz) to test the robustness of the results. Both the reviewer and the authors have a point here. I don't think there is a unique solution to this problem though, best to report both.

We thank reviewer 5 for his recommendation. We repeated the calculations of both models (MDD versus HC and Non-Responders versus Responders) after applying a low-pass filter of 30Hz to exclude potential sources of muscle artefacts. Results revealed slightly lower accuracies across the settings. We included a short report on these findings and a few sentences to discuss the results in the main text and give more extensive information in the supplement.

Main text (method section):

“(Additional analysis on a 0.5-30Hz band pass filter can be found in the supplement).”

Main text (result section):

“A slightly decreased accuracy was also found when applying a 30Hz low-pass filter instead of 45Hz low pass filter (see supplement) with 65%.”

And

“A decreased accuracy of 55% was also found when applying a 30Hz low-pass filter instead of 45Hz low pass filter (see supplement).”

Main text (discussion section):

“Moreover, applying a more stringent low-pass filter at 30 Hz instead of 45 Hz resulted in reduced classification accuracies across settings. It remains unclear whether this decrease reflects the removal of informative but potentially muscle-related electrophysiological activity, or whether genuine EEG gamma-band activity in the 30–45 Hz range contributes to the improved performance observed when retaining these higher frequencies.”

Supplement (1.11 Differences of eye-related movements in the EEG data across the groups):

“Classification of MDD subjects versus HC subjects from all datasets showed a train accuracy of 70.92% (compared to 73.16% with a 45Hz filter) and a validation accuracy of 68.0% (compared to 71.13% with a 45Hz filter). Test accuracy for all pooled segments of unseen subjects from the test set was 63.44% (compared to 64.84% with a 45Hz filter) in total (not subject-wise) and the percentage of correctly classified unseen subjects based on the number of correct classified segments of a subject was 65% (compared to 67.5% with 45Hz filter).

Classification of responders versus non-responders of SSRI treatment showed 69.94% (versus 71.8% for 45Hz filter) accuracy for single segments for the training set after 708 training epochs. Validation set accuracy was 65.42% (versus 66.4% in the 45Hz filter model). The test set reached 59% (64% for 45Hz filter) for pooled segments and a mean value of 55% (against 79% for 45Hz filter) for correct subject-wise classification.

Thus, all results were slightly worse when applying a low-pass filter of 30Hz instead of 45Hz. Although it cannot be ruled out that muscle activity in the range from 30-45Hz contributes to the correct labelling, another explanation would be that EEG-gamma activity in this range yields important information for a correct classification. Since answering this question was beyond the scope of the presented work, further studies on this topic seem necessary.”

R5.2 Point 3.2 I think this is a valid point of the reviewer. The correction method they used is fairly established and if the classification performance drops it indeed could be the case that the classifier partially relies on artefacts. There may also be a loss of power. It is true that the responder/non-responder classification includes two patient groups and thus the explanation of the loss of accuracy cannot simply be patients blink more, but it may still be that responders blink more than non-responders for example. A simple and more convincing test to rule this out would be to train a classifier purely on the EOG data and see how well it performs or even to simply test for differences in the number of blinks or artefacts across both groups.

We appreciate the comment of reviewer 5. We agree that this topic needs more clarification and thus decided to do additional analysis, following the recommendation of reviewer 5. Since a one-dimensional signal (only EOG activity) did not yield enough data for a deep-learning approach, we decided to apply the second suggestion of reviewer 5 and tested the numbers of artefacts derived from eye-activity for the different groups. No differences between MDD and HC were found nor between Responders and Non-Responders. We decided to include this information in the main text and give more extensive results in the supplement:

Main text (methods section):

“Further, we conducted a comparison of the amount of eye-related events/minute for all datasets (details in supplement).”

Main text (results section):

“No differences in eye-related movements were found between groups (supplement).”

and

“No differences in eye-related movements were found between responders and non-responders (supplement).”

Main text (discussion section):

“To rule out potential confounding effects of group differences in eye-movement-related events, we quantified their occurrence across all settings and found no significant differences between patients and healthy controls, nor between responders and non-responders (see supplement).”

Supplement (1.12 Differences of eye-related movements in the EEG data across the groups):

“To determine whether the number of eye-derived artefacts was different between the tested groups and thus may have contributed to the classification results, additional analysis was performed. Eye blinks were automatically identified using the MNE-Python function `find_eog_events`. The EEG was band-pass filtered between 1 and 8 Hz to isolate the slow ocular deflections characteristic of blinks. A bipolar channel based on electrooculogram (EOG) data or reconstructed EOG channels (F7-F8) served as an electrooculographic (EOG) surrogate. Peaks exceeding an amplitude threshold of 150 μV were classified as blink events and used for subsequent quantification (blinks per minute). The amount of eye-related movement events per minute for all groups (HC versus MDD and Responders versus Non-Responders) were then tested against each other using a t-test. No differences between MDD and HC were found nor between Responders and Non-Responders.

Table 1: Given are the mean numbers of eye-movement events per minute, standard deviation for the different groups.

Group	Mean	SD(%)
HC	0.26	1.12%
MDD	0.35	1.47%
Responders	0.40	1.91%
Non-Responders	0.33	0.99%

Table 2: Results of the comparison between groups for eye-movement related events across the different labelling groups. There were no significant differences between MDD and HC and between Responders and Non-Responders.”

Comparison	t-value	p-value
HC vs MDD	-0.69	0.49
Responders vs Non-Responders	0.34	0.74

R5.3 My recommendation would be to repeat the analyses with a 30 Hz low-pass cutoff in the supplement and test for the differences in the number of artefacts or blinks across groups or train a classifier on the EOG data to test how well one can do on eyeblinks alone. This would fully address the concerns that were in my opinion rightfully raised by the reviewer.

We want to thank reviewer 5 for his contributions. We added the suggested calculations and reported and discussed on them. We think that these new analyses strengthen the results of the presented work. We hope that adding these additional analysis (repeating the deep-learning model training with data with low-pass filter at 30Hz and comparing eye-movement related activities between the groups) now meets the expectations.